# The Effects of Interaction Scenarios on EFL Learners’ Technology Acceptance and Willingness to Communicate with AI

**DOI:** 10.3390/bs15101391

**Published:** 2025-10-14

**Authors:** Zheng Cui, Hua Yang, Hao Xu

**Affiliations:** 1School of Foreign Studies, Northeastern University, Shenyang 110819, China; cuizheng@neuq.edu.cn; 2School of English for Specific Purposes, Beijing Foreign Studies University, Beijing 100089, China; 3National Research Centre for Foreign Language Education, Beijing Foreign Studies University, Beijing 100089, China; xuhaokent@bfsu.edu.cn

**Keywords:** sociocultural theory, AI interaction scenarios, technology acceptance (TA), willingness to communicate with AI (AI-WTC), EFL learning

## Abstract

Grounded in a sociocultural theory, this study investigates how distinct interaction scenarios influence Chinese English as a Foreign Language (EFL) learners’ technology acceptance: perceived usefulness (PU) and perceived ease of use (PEU), and their willingness to communicate with AI (AI-WTC). A total of 367 university students completed a scenario-based questionnaire measuring PU, PEU, and AI-WTC across four empirically derived scenarios: advisory interaction, language skills support, academic knowledge inquiry, and factual information retrieval. Repeated-measures ANOVA with Bonferroni tests revealed significant scenario effects on all three constructs, though effect sizes were small to moderate. Factual Information Retrieval Scenario consistently received the highest ratings, whereas Academic Knowledge Inquiry and Language Skills Support Scenario scored lowest. A salient divergence emerged in complex scenarios: Advisory Interaction Scenario was rated more useful than Language Skills Support Scenario, yet both elicited equally low willingness to communicate, indicating that perceived usefulness alone may not sustain engagement under high interactional demands. These findings suggest that the effectiveness of AI as a communicative scaffold is not inherent but co-constructed through scenario-specific affordances and constraints. The study contributes a scenario-sensitive framework to TAM and WTC research, providing pedagogical guidance for designing differentiated AI-mediated language tasks.

## 1. Introduction

The rapid integration of generative artificial intelligence (GenAI) into educational landscapes has marked a pivotal moment for foreign language pedagogy ([13]; [20]; [6]). Advanced AI tools, such as ChatGPT and other large language models, are no longer mere supplementary instruments but are increasingly positioned as active participants in the learning process. This paradigm shift prompts a reconceptualization of learner–computer interaction ([27], [28]; [37]), wherein, from a sociocultural perspective, GenAI transcends its role as a simple tool to become a “communicative agent” ([12]; [42]). This emerging dynamic presents both unprecedented opportunities and notable challenges for second language (L2) education ([2]; [18]; [4]). While GenAI offers personalized, on-demand interaction that can mitigate foreign language anxiety and provide extensive output practice, its pedagogical effectiveness is not guaranteed, as empirical findings remain mixed ([11]; [17]; [46]). The variability in outcomes underscores that AI’s potential is not self-actualizing but is contingent upon learners’ psychological and behavioral engagement. Consequently, the extent to which learners accept this technology (Technology Acceptance) and their willingness to communicate with a non-human agent (WTC) emerge as critical factors that will ultimately determine the instructional value of GenAI ([8]; [45]).

The Technology Acceptance Model (TAM) has been the cornerstone of research on user adoption of new technologies for over three decades ([3]). Developed by [8] ([8]), the original TAM proposed that a user’s intention to adopt a technology is primarily determined by two key beliefs: perceived usefulness (PU) and perceived ease of use (PEU). As [8] ([8]) defines them, PU refers to “the degree to which a person believes that using a particular system would enhance his or her job performance,” while PEU is “the degree to which a person believes that using a particular system would be free of effort.” These two factors influence a user’s attitude toward using the system, which in turn predicts their behavioral intention and actual use. When AI functions as a communicative agent, PU and PEU are no longer confined to the tool’s interface or functional output; they become intricately linked to the perceived quality of the interaction itself ([38]). Recent studies applying TAM to ChatGPT in educational contexts have confirmed the continued importance of PU and PEU in shaping learners’ acceptance of AI-based tools ([27]; [26]; [19]; [34]; [42]).

Beyond technology adoption, learners’ communicative engagement with AI also hinges on affective and psychological readiness, as explained by Willingness to Communicate (WTC) theory. WTC describes an individual’s readiness to engage in discourse ([31]; [29]) and has been widely applied in second language acquisition research. In the L2 context, WTC is recognized not merely as a stable trait but as a dynamic, situation-specific construct that is susceptible to a complex interplay of linguistic, affective, and situational variables ([32]). With the rise of digital technologies, L2 WTC research has expanded into computer-mediated communication ([23]). Early findings not only validate the applicability of the WTC framework to AI contexts but are also refining its contours, revealing critical distinctions; while AI can offer a low-anxiety space ([24], [25]; [44]), learners’ willingness to engage is profoundly shaped by their perception of the AI’s competence, reliability, and the specific purpose of the interaction ([11]; [42]). This evolving line of inquiry not only demonstrates the adaptability of the WTC framework to AI-mediated ecologies but also highlights that AI-WTC is a unique construct sensitive to technological and contextual affordances.

Although TA and AI-WTC have been explored in related strands of research, studies that examine both constructs within the same AI interaction scenario remain limited. This gap is compounded by a tendency in prior research to treat AI as a monolithic entity, thereby overlooking the fundamental ways in which specific interaction scenarios shape learner perceptions and behaviors. This limitation points to the need for a more differentiated analytical framework that can capture the variability of AI interaction scenarios, a perspective reinforced by emerging studies showing that learners engage with AI in distinct, role-based capacities ([35]). In this light, a scenario-sensitive perspective offers a promising direction for understanding how PU, PEU and AI-WTC interact in more nuanced ways.

In summary, the existing literature reveals three critical gaps that this study aims to fill. First, while WTC theory is being extended to AI, research has not yet systematically examined how different types of interaction with AI influence learners’ communicative volition. Second, while the TA model is being applied to GenAI, studies have largely overlooked how the purpose of use (i.e., the scenario) dynamically shapes learners’ perceptions of usefulness and ease of use. Third, and most importantly, existing research on learner–AI interaction, though expanding, tends to conceptualize it as a uniform activity. Most studies examine “the effect of ChatGPT” or “the impact of AI” (e.g., [11]; [26]) without differentiating how distinct purposes or scenarios of interaction may shape learning outcomes ([43]). This assumption of uniformity obscures the contextual nature of AI-mediated communication and hinders the formulation of targeted pedagogical strategies.

Drawing on sociocultural theory, this study focuses on Chinese EFL learners, examining how scenarios shape their technology acceptance and willingness to communicate with AI. Scenarios define the communicative event by shaping the roles, expectations, and goals assigned to the learner and the AI, thereby influencing both technology acceptance and communicative engagement. Within this framework, learners’ technology acceptance, operationalized as PU and PEU, is expected to vary across four pedagogically relevant interaction scenarios: advisory interaction (AIS), language skills support (LSSS), academic knowledge inquiry (AKIS), and factual information retrieval (FIRS). Similarly, learners’ AI-WTC is theorized to be scenario-sensitive, reflecting the extent to which contextual affordances lower interactional barriers and foster readiness for dialogue. Finally, we examine the extent to which scenario effects on TA correspond to scenario effects on AI-WTC, thereby revealing potential interconnections between learners’ cognitive evaluations of AI and their communicative volition (See Figure 1 for the conceptual model).

Guided by this framework, the study addresses three research questions:

(1) To what extent do interaction scenarios affect learners’ technology acceptance (PU and PEU)?

(2) To what extent do interaction scenarios affect learners’ willingness to communicate with AI (AI-WTC)?

(3) What are the potential interconnections between the effects of interaction scenarios on technology acceptance and WTC?

Based on this, we propose the following hypotheses:

**H1.** 
*Learners’ technology acceptance, measured as PU and PEU, will significantly differ across the four interaction scenarios: Advisory Interaction Scenario, Language Skills Support Scenario, Academic Knowledge Inquiry Scenario, and Factual Information Retrieval Scenario.*


**H2.** 
*Learners’ willingness to communicate with AI (AI-WTC) will significantly vary across the four scenarios.*


**H3.** 
*Scenario-based differences in technology acceptance (PU and PEU) will correspond to scenario-based differences in AI-WTC.*


## 2. Materials and Methods

### 2.1. Participants and Procedures

The participants were 367 Chinese university students recruited from two key universities in northern China. The sample comprised both undergraduate and graduate students, with a mean age of 19.93 years. In terms of academic background, 71 students (19.35%) majored in English, 79 (21.53%) in other foreign languages, and 217 (59.13%) in non-language disciplines. Regarding academic level, 233 (63.49%) were first-year undergraduates, 61 (16.62%) were second-year, 33 (8.99%) were third-year, 10 (2.72%) were in their final undergraduate year, 17 (4.63%) were master’s students, and 1 participant (0.27%) was a doctoral student. All participants had prior experience using generative AI tools (e.g., ChatGPT and Wenxin Yiyan) for auxiliary language learning purposes, as the use of such technology is increasingly integrated into the general academic environment at their institutions. The study specifically focused on the context of learning English as a foreign language, and the diverse background ensured a wide range of English proficiency levels and learning needs, thereby enhancing the representativeness of the findings for the broader EFL learner population in Chinese higher education.

Given the lack of existing instruments measuring technology acceptance and willingness to communicate with AI across context-specific interaction scenarios, the present study developed a purpose-designed questionnaire through a multi-phase process. For the TA scale, we adopted [8]’s ([8]) Technology Acceptance Model (TAM) as the theoretical framework, adapting items from the original TAM questionnaire ([8]) to fit AI-mediated language learning contexts. The TA scale comprised two subscales: Perceived Usefulness, with six items assessing learners’ beliefs about AI’s effectiveness in enhancing learning outcomes, and Perceived Ease of Use (PEU), with six items evaluating the accessibility of AI interactions. All items were designed to be scenario-adaptive, allowing for later contextualization within specific interaction scenarios. The AI-WTC scale was adapted from [36]’s ([36]) classroom-based WTC questionnaire, which highlights the interplay between individual and contextual variables. Building on this foundation, the present study extended the individual dimension to include learners’ technology acceptance alongside their AI-WTC. With communicative engagement, AI was understood as co-constructed through the interaction between learners’ perceptions and scenario-specific affordances. To capture this, the AI-WTC scale employed an “I am willing to …” structure across four scenarios, with six items per scenario developed in parallel with the TA scale to ensure measurement consistency.

With the theoretical space for TA and WTC specified, we adopted a theory-driven, empirically calibrated approach to scenario construction. Drawing a priori on [35]’s ([35]) taxonomy of AI roles in L2 learning, we operationalized four interaction scenarios: Advisory Interaction Scenario, Language Skills Support Scenario, Academic Knowledge Inquiry Scenario, and Factual Information Retrieval Scenario, to delineate the content domain in which AI assumes distinct social-interactional roles. To develop ecologically valid scenarios for the subsequent scale, we first administered an open-ended questionnaire to a separate, independent sample of 67 university-level EFL learners ([16]), eliciting detailed accounts of their actual and envisioned AI-supported learning activities (938 words). The qualitative data underwent a three-phase thematic analysis involving initial open coding to identify instances of AI use, categorization of codes into broader themes, and consolidation into four overarching scenarios. This process yielded 130 concrete interaction instances. Six representative tasks were selected per scenario based on frequency and pedagogical salience ([10]), with each task either reflecting experiences frequently reported in the open-ended responses or clearly described with sufficient context to ensure participants’ familiarity, so that students could meaningfully evaluate them even if they had not previously performed a specific task.

Building upon the empirically validated scenario framework, we systematically integrated the 24 interaction tasks into the measurement instrument by contextualizing all questionnaire items for technology acceptance (PU/PEU) and AI-WTC within specific scenario parameters. Each of the six tasks per scenario type was carefully embedded into parallel item sets for both constructs, ensuring that respondents evaluated identical interaction contexts across all measures. This dual contextualization process yielded a 72-item instrument (24 scenarios × 3 constructs), enabling direct comparisons of how specific scenario characteristics influence both technology acceptance and communicative willingness.

To ensure that participants could respond without language barriers, the original English questionnaire was collaboratively translated into Chinese by the research team. Semantic and cultural equivalence was maintained through iterative reviews ([5]). The preliminary Chinese version was then administered to 40 first-year undergraduate students, who provided feedback on item clarity and relevance. Subsequently, two experts in applied linguistics and educational technology reviewed the revised instrument, offering suggestions to further enhance construct validity and alignment with the intended theoretical dimensions ([9]). Based on this feedback, the final version of the questionnaire was produced.

The finalized survey was distributed online via Wenjuanxing (https://www.wjx.cn/), a widely used data collection platform in China. A simple random sampling method was employed, and data were collected from 13 July to 24 July 2025. Ethical considerations were strictly observed: participants were informed of the study’s purpose, assured of voluntary participation, and notified of their right to withdraw at any time. No personally identifiable information was collected, and submission of the completed survey was considered informed consent. On average, respondents completed the questionnaire in approximately 6 min.

### 2.2. Instrument

The instrument of this study was an online survey comprising two main sections. The first section collected participants’ demographic information, including gender, age, academic major, and year or level of study (based on the 2024–2025 academic calendar). This information was used to describe the sample and control for potential background variability.

The second section consisted of two scenario-based scales designed to assess learners’ perceptions of AI and their communicative willingness with it. Four distinct AI interaction scenarios were identified to reflect the range of communicative events in which second language learners may engage with AI tools. The four empirically derived interaction scenarios are defined as follows: the Advisory Interaction Scenario involves seeking personalized advice or strategic feedback; the Language Skills Support Scenario focuses on practicing or refining specific linguistic abilities; the Academic Knowledge Inquiry Scenario entails accessing explanatory theoretical or disciplinary knowledge; and the Factual Information Retrieval Scenario concerns retrieving concise, referential information. These scenarios form the contextual foundation for measuring TA and AI-WTC. The first scale targeted TA, measuring two constructs across four interaction scenarios: Perceived Usefulness, referring to the extent to which learners believe that using AI tools would enhance their foreign language learning outcomes. Perceived Ease of Use (PEU) refers to the degree to which learners perceive AI tools as easy and accessible for learning-related tasks. The second scale measured learners’ AI-WTC across four scenarios. In this study, AI-WTC was reconceptualized to capture learners’ context-sensitive readiness to initiate or engage in communicative interactions with AI tools in the context of L2 learning.

The structure and item distribution of the scenario-based questionnaire are presented in Table 1. All items were measured using a five-point Likert scale, ranging from 1 (strongly disagree) to 5 (strongly agree). The full list of items, including construct-scenario mappings, is provided in Appendix A.

Measurement reliability and construct validity were assessed using AMOS 26, where all standardized factor loadings exceeded 0.6 and were statistically significant (*p* < 0.05), indicating strong associations between each item and its respective construct and a well-fitting measurement model. Composite reliability (CR) values ranged from 0.897 to 0.955, exceeding the recommended threshold of 0.70 and demonstrating strong internal consistency. Average variance extracted (AVE) values ranged from 0.593 to 0.779, surpassing the minimum requirement of 0.50 and supporting convergent validity (see Appendix B, Table A1). Inter-factor correlations, calculated using SPSS 26, ranged from 0.70 to 0.91 between PU and PEU, and from 0.72 to 0.88 between TA and AI-WTC (see Appendix B, Table A2). Although correlations were relatively high, this pattern aligns with theoretical expectations given the close conceptual relationships among constructs. Overall, these results indicate that the instrument provides a reliable and valid basis ([33]) for examining technology acceptance and willingness to communicate with AI.

### 2.3. Data Analysis

A quantitative research approach was adopted to examine how learners’ technology acceptance and willingness to communicate with AI tools vary across distinct interaction scenarios in L2 learning. All statistical analyses were conducted using IBM SPSS Statistics (version 26.0). The data analysis proceeded in three stages, each aligned with one or more of the study’s research questions.

To address the first and second research questions, namely, whether learners’ PU, PEU, and AI-WTC differ significantly across distinct interaction scenarios, a repeated measures analysis of variance (RM-ANOVA) was employed ([14]; [30]). Separate RM-ANOVAs were conducted for each of the three constructs, as they were all measured repeatedly across the same four within-subject conditions. This sequential analytical approach enabled the detection of within-person variability while controlling for individual differences in baseline responses, which is a key strength of within-subject designs ([15]).

Following this, to answer the third research question, whether scenario-based differences in TA align with variations in AI-WTC, post hoc pairwise comparisons were performed using the Bonferroni adjustment method ([1]). This conservative correction was used to minimize the risk of inflated Type I error associated with multiple comparisons. The aim was to identify potential convergence or divergence between learners’ evaluations of AI and their willingness to engage with AI across different interactional contexts.

## 3. Results

### 3.1. Scenario-Based Differences in Technology Acceptance (PU and PEU)

To examine whether learners’ technology acceptance, operationalized as Perceived Usefulness and Perceived Ease of Use, varied significantly across the four AI interaction scenarios, a within-subjects design was employed, as the same group of participants provided ratings for each construct under all four scenario conditions: Advisory Interaction Scenario, Language Skills Support Scenario, Academic Knowledge Inquiry Scenario, and Factual Information Retrieval Scenario. Repeated Measures Analysis of Variance (RM-ANOVA) was conducted using IBM SPSS Statistics (version 26.0) to account for the correlated nature of the data and to assess within-subject variability while controlling for individual differences in baseline perceptions.

The results, presented in Table 2, revealed a statistically significant main effect of interaction scenario on PU, F(3, 1098) = 16.096, *p* < 0.001, η^2^ = 0.042, and on PEU, F(3, 1098) = 24.641, *p* < 0.001, η^2^ = 0.063. According to [7]’s ([7]) benchmarks, these effect sizes indicate small-to-moderate effects, suggesting that while the differences are statistically robust, scenario-related differences in technology acceptance account for only a modest proportion of the total variance. This indicates that learners’ perceptions of AI’s usefulness and ease of use varied across scenarios.

### 3.2. Scenario-Based Differences in Willingness to Communicate with AI

To examine potential differences in learners’ willingness to communicate with AI across the four interaction scenarios, a repeated measures analysis of variance (RM-ANOVA) was performed. This approach enabled comparison of WTC scores for the same participants in the Advisory Interaction Scenario, Language Skills Support Scenario, Academic Knowledge Inquiry Scenario, and Factual Information Retrieval Scenario, while accounting for individual variability.

As shown in Table 3, the analysis identified a significant main effect of interaction scenario on AI-WTC, F(3, 1098) = 9.600, *p* < 0.001, η^2^ = 0.026. Interpreted against [7]’s ([7]) guidelines, this effect size is small, indicating that scenario type exerts a measurable yet limited influence on learners’ communicative willingness with AI. In other words, although learners’ AI-WTC scores varied across scenarios, the extent of these differences represented only a modest share of the overall variance.

### 3.3. Interconnections Between Technology Acceptance and AI-WTC Across Scenarios

To further identify how TA and AI-WTC were interrelated across the four interaction scenarios, Bonferroni-adjusted post hoc pairwise comparisons were conducted following the RM-ANOVA results. This procedure allowed for the identification of specific scenario pairs that differed significantly, while controlling for the inflated Type I error rate associated with multiple comparisons ([1]).

As displayed in Table 4, the post hoc analyses revealed variations in scenario effects across the constructs. For PU, the overall effect of scenario type was significant, F(1) = 18.823, *p* < 0.001, η^2^ = 0.049, indicating that a small proportion of the variance in perceptions of usefulness was accounted for by the scenario type. Pairwise comparisons suggested a pattern where the Factual Information Retrieval Scenario received the highest ratings, significantly surpassing all others. The Academic Knowledge Inquiry Scenario followed, while the Language Skills Support Scenario received the lowest ratings. Notably, the Advisory Interaction Scenario was perceived as more useful than the Language Skills Support Scenario but less useful than the Factual Information Retrieval Scenario. For PEU, a significant main effect was observed with a moderate-to-large effect size, F(1) = 51.774, *p* < 0.001, η^2^ = 0.124, indicating that scenario type accounted for a more substantial portion of the variance in ease-of-use perceptions. Bonferroni comparisons indicated that the Factual Information Retrieval Scenario again received the highest ratings, significantly exceeding all other scenarios, while the Advisory Interaction Scenario consistently ranked lowest. The relative positioning of the Language Skills Support Scenario and the Academic Knowledge Inquiry Scenario (LSSS < AKIS < FIRS) suggested a potential gradient related to task complexity. For AI-WTC, a statistically significant main effect was observed, F(1) = 22.903, *p* < 0.001, η^2^ = 0.059, representing a small-to-moderate effect size where scenario type explained a limited portion of the variance in willingness to communicate. Post hoc comparisons indicated a pattern: the Factual Information Retrieval Scenario elicited the highest levels of AI-WTC, significantly surpassing other scenarios. The Academic Knowledge Inquiry Scenario occupied an intermediate position, generating significantly greater AI-WTC than both the Advisory Interaction Scenario and the Language Skills Support Scenario, between which no significant difference was found.

Collectively, these patterns indicate scenario-dependent variation, with the Factual Information Retrieval Scenario consistently rated highest and the Advisory Interaction Scenario (AIS) or Language Skills Support Scenario lowest. The nuanced divergences reveal several insights. First, the effect size for PEU (η^2^ = 0.124) was notably larger than for PU (η^2^ = 0.049) and AI-WTC (η^2^ = 0.059), suggesting that judgments of ease of use were more sensitive to contextual differences than those of usefulness or willingness to communicate. Second, the disparity between the Advisory Interaction Scenario and the Language Skills Support Scenario in PU (AIS > LSSS) versus their equivalence in AI-WTC (AIS ≈ LSSS) might tentatively suggest that perceived utility alone does not necessarily translate to a higher AI-WTC in complex interaction scenarios. Finally, the varying gap between the Factual Information Retrieval Scenario and the Academic Knowledge Inquiry Scenario across constructs highlights differential sensitivity to task structure. These findings collectively showcase that AI-mediated communication is shaped by, but not reducible to, traditional technology acceptance mechanisms.

## 4. Discussion

The present study aimed to investigate how distinct AI interaction scenarios influence EFL learners’ technology acceptance and their willingness to communicate. Regarding the first research question, the findings demonstrated that learners’ perceptions of usefulness and ease of use differed significantly across the four scenarios, thereby supporting Hypothesis 1. However, the effect sizes were small to moderate ([7]), indicating that although scenario type exerted a reliable influence, it accounted for only a limited proportion of the overall variance. From a sociocultural perspective, this pattern suggests that learners’ evaluation of AI technology is not determined solely by the inherent affordances of the tool but also shaped by the situated communicative activity in which AI is embedded ([22]; [41]). The modest yet significant scenario effects point to the role of cultural factors in the Chinese learning context, where the inherent complexity of different tasks creates varying cognitive burdens that subsequently shape learners’ perceptions of the technology’s usefulness and ease of use.

With respect to the second research question, the results showcased that learners’ AI-WTC also varied across the four scenarios, thereby confirming Hypothesis 2. Similar to the findings for technology acceptance, the effect sizes were small to moderate, suggesting that although interaction scenarios exert a consistent influence, the variance they explain remains modest. This finding aligns with sociocultural perspectives that conceptualize WTC not as a fixed trait but as a context-sensitive construct, co-constructed through situated interaction and shaped by the cultural background embedded in each scenario ([29]; [21]).

The third research question examined the interconnections between scenario effects on technology acceptance and willingness to communicate with AI. The results showcase a clear hierarchy: the Factual Information Retrieval Scenario consistently secured the highest ratings across all constructs, closely followed by the Academic Knowledge Inquiry Scenario. This stable alignment indicates that for tasks with low complexity and a clear goal, akin to the structured exercises prevalent in Chinese foreign language education, AI’s scaffolding as a “more capable peer” ([41]) operates effectively within the learner’s ZPD. The predictability and minimal communicative risk of these scenarios lower cognitive load, thereby supporting both positive technology acceptance and communicative engagement. In contrast, a more complex pattern emerges at the lower end of the spectrum. Although both the Advisory Interaction Scenario and the Language Skills Support Scenario received the poorest ratings, their internal profiles diverged: the former was perceived as more useful yet harder to use, while both ultimately led to equally low willingness to communicate. The modest effect sizes affirm that this dissociation, while systematic, is one of several factors at play. This pattern can be interpreted by examining the distinct challenges each scenario poses. The Language Skills Support Scenario demands high linguistic precision, creating a communicative risk of evaluation. Unlike a human teacher who can offer adaptive support, the AI’s standardized feedback fails to provide the nuanced scaffolding needed to mitigate this risk, thus falling short as an accessible “more capable peer” within the learner’s ZPD ([39]). The Advisory Interaction Scenario, conversely, imposes a high cognitive load from strategic judgment and open-ended information processing ([40]). In both cases, the AI’s scaffolding is either perceived as inadequate for the task or too demanding to integrate effectively. Consequently, compared to the highly accessible support in the Factual Information Retrieval Scenario, these complex scenarios render the AI a less effective peer, suppressing willingness to communicate despite potential utility.

This study makes threefold contributions. At the theoretical level, it moves beyond a monolithic view of AI by introducing a scenario-sensitive perspective into the Technology Acceptance Model and Willingness to Communicate framework. It shows that the relationship between technology acceptance and communication willingness is not fixed but shifts according to the alignment between AI’s affordances and the demands of particular interaction scenarios. This reframing shifts attention from evaluating the technology in isolation to understanding how usage contexts shape learner perceptions and behaviors. By drawing on sociocultural theory, the study further conceptualizes AI as a “more capable peer” whose effectiveness depends on scenario-mediated scaffolding within the learner’s ZPD. Methodologically, the study develops and validates a scenario-based instrument that captures the nuances of human–AI interaction across diverse contexts, providing a resource for future empirical investigations. Practically, the findings offer educators an evidence-based framework for differentiated AI task design and sequencing, while also encouraging designers to build adaptive scaffolding mechanisms that lower interactional friction and promote learner engagement. Together, these contributions advance the understanding of AI-mediated communication by foregrounding the critical, yet underexplored, role of interaction scenarios.

## 5. Conclusions

Grounded in the sociocultural theory of viewing AI as a “more capable peer” ([41]), this study quantitatively investigated how four distinct AI interaction scenarios shape Chinese EFL learners’ technology acceptance and their AI-WTC. The results demonstrated that scenario type exerts a significant, albeit modest, influence on both constructs. A generally aligned yet nuanced pattern was observed: while the Factual Information Retrieval Scenario was consistently rated highest across all measures, the relative positioning of the Advisory Interaction and Language Skills Support scenarios differed between constructs. A key finding was the critical divergence in complex scenarios: despite being rated as more useful than language practice, the advisory scenario triggered an equally low level of communicative willingness, suggesting that high perceived usefulness alone is insufficient to foster engagement when interactional demands are high.

The findings highlight that effective AI integration in language education requires intentional scenario design aligned with both technological capabilities and pedagogical goals. Importantly, since most AI systems were not originally designed as language teachers but rather as general-purpose conversational agents, expectations need to be recalibrated by positioning AI primarily as a scaffolding tool or a “more capable peer” rather than a comprehensive teacher substitute. This reframing acknowledges that AI’s core strengths lie in efficient information processing and structured support, whereas nuanced guidance and empathy remain the domain of human educators, especially in complex advisory roles. Furthermore, while the observed effects of scenario type were small to moderate, they nonetheless indicate that sequencing tasks from high to low AI–compatibility may be a prudent pedagogical strategy. Specifically, structured scenarios such as Factual Information Retrieval and Academic Knowledge Inquiry can serve as accessible entry points, leveraging AI’s informational strengths to build learner confidence through low-risk, high-success interactions. As learners gain proficiency, more open-ended scenarios such as Language Skills Support and Advisory Interaction can be gradually introduced. At this stage, explicit scaffolding becomes crucial: curated prompt templates (e.g., “Act as a critical friend and evaluate the thesis statement of my essay…”) and metacognitive training can guide learners to engage productively with AI while critically evaluating its feedback. For highly complex scenarios like Advisory Interaction, strategically permitting the use of the native language can serve as a transitional scaffold that reduces cognitive load, allowing learners to focus on complex advisory tasks before transitioning to the target language. Finally, hybrid human–AI models offer an effective balance, employing AI for initial support while reserving human oversight for higher-order tasks such as assessing argument coherence and cultural appropriateness. Framed through [41]’s ([41]) notion of scaffolding, this developmental sequencing illustrates how AI’s role can evolve from an authoritative knowledge resource to a collaborative partner, enabling educators to maximize its utility while mindfully addressing its limitations.

Several limitations in this study warrant consideration. First, the reliance on self-reported data without behavioral triangulation may not fully capture actual engagement, as responses could be influenced by social desirability bias. Second, the generalizability is limited by the Chinese student sample, as culturally specific factors such as the pronounced novelty effect and a tendency toward social desirability in educational settings may have uniquely shaped the responses. Third, the static measurement of technology acceptance and AI-WTC at a single time point overlooks how these perceptions might evolve with prolonged AI use. Future research could address these limitations through several avenues. First, behavioral data such as chat logs or AI interaction transcripts could be analyzed to triangulate learners’ self-reported perceptions with actual communicative behavior, offering richer evidence of engagement patterns. Second, cross-cultural comparisons involving learners from different sociocultural contexts would help clarify whether the observed scenario effects are culturally bound or generalizable across settings. Third, longitudinal studies could trace how learners’ technology acceptance and AI-WTC evolve over extended periods of interaction, revealing potential developmental trajectories of AI-mediated communication in L2 learning.

## Figures and Tables

**Figure 1 behavsci-15-01391-f001:**
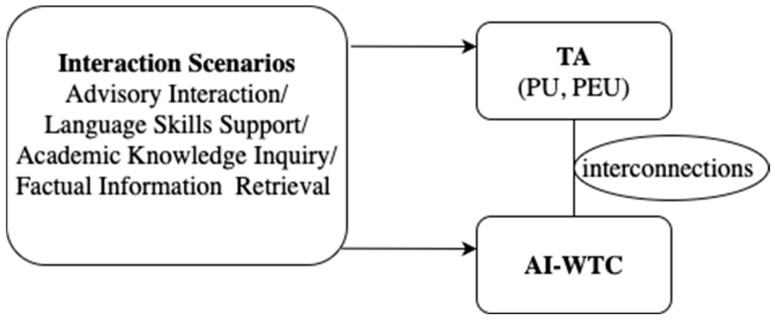
Conceptual Model.

**Table 1 behavsci-15-01391-t001:** Structure and item distribution of the scenario-based questionnaire.

Section	Content/Construct	Scenarios	No. of Items
1. Background Information	Gender, Age, Major, Academic Year		4
2. A. TA	PU, PEU	AIS, LSSS, AKIS, FIRS	48
B. AI-WTC	AI-WTC	AIS, LSSS, AKIS, FIRS	24
		Total	72

Note. PU = Perceived Usefulness; PEU = Perceived Ease of Use; AI-WTC = Willingness to Communicate with AI. AIS = Advisory Interaction Scenario; LSSS = Language Skills Support Scenario; AKIS = Academic Knowledge Inquiry Scenario; FIRS = Factual Information Retrieval Scenario.

**Table 2 behavsci-15-01391-t002:** Results of repeated measures ANOVA for interaction scenario effects on PU, PEU.

Dependent Variable	F	df	*p*	η^2^
PU	16.096	3, 1098	<0.001	0.042
PEU	24.641	3, 1098	<0.001	0.063

Note. PU = Perceived Usefulness; PEU = Perceived Ease of Use.

**Table 3 behavsci-15-01391-t003:** Results of Repeated Measures ANOVA for Interaction Scenario Effects AI-WTC.

Dependent Variable	F	df	*p*	η^2^
AI-WTC	9.600	3, 1098	<0.001	0.026

Note. AI-WTC = Willingness to Communicate with AI.

**Table 4 behavsci-15-01391-t004:** Bonferroni Post Hoc Comparisons across Interaction Scenarios for PU, PEU, and WTC.

Variable	F	df	*p*	η^2^	Comparisons
PU	18.823	1	0.000	0.049	FIRS > AKIS > AIS > LSSS
PEU	51.774	1	0.000	0.124	FIRS > AKIS > LSSS > AIS
AI-WTC	22.903	1	0.000	0.059	FIRS > AKIS > LSSS ≈ AIS

Note. PU = Perceived Usefulness; PEU = Perceived Ease of Use; AI-WTC = Willingness to Communicate with AI; AIS = Advisory Interaction Scenario; LSSS = Language Skills Support Scenario; AKIS = Academic Knowledge Inquiry Scenario; FIRS = Factual Information Retrieval Scenario.

## Data Availability

A publicly available dataset was analyzed in this study. This data can be found here: https://pan.baidu.com/s/19g82JL2jYQDqc0ArEFip9g?pwd=data (accessed on 22 August 2025) (password: data).

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
