# Peer review of "The Effects of Interaction Scenarios on EFL Learners’ Technology Acceptance and Willingness to Communicate with AI"

_behavsci, 2025, doi:10.3390/bs15101391_

Round 1
Reviewer 1 Report
Comments and Suggestions for Authors
This study examined the effects of four AI interaction scenarios on L2 learners’ technology acceptance (i.e., perceived usefulness and perceived ease of use) and their willingness to communicate with AI with university students. The authors have clearly identified the research gaps supported by literature review. While the research focused solely on quantitative method, the validation of the questionnaire was not clearly presented. Robust validation is critical as questionnaire items were newly developed.
Detailed comments are as follows:
- Abstract and Title
- The title and abstract are appropriate. They reflect the main theme of the paper and provide readers with a sense of research focus
- Introduction
- Generally, the introduction is clear and well structured. The authors contextualized research problems within the broader literature, demonstrating why the topic was relevant. Key terms were defined, the flow from general background to specific research gap was coherent and the rationale for conducting the study was articulated with clarity.
- Learner-AI interaction is not novel (lines 42-44). The claim that this interaction is novel was not supported with reference. The claim was also repeated in lines 45-46.
- The argument (lines 52-55) about learner acceptance and willingness to communicate in determining the value of AI appeared rather abruptly.
- Research question 2 starting with ‘how’ is more appropriate for qualitative study. Since this study focused solely on quantitative, quantitative research questions are critical, for example. “to what extent do scenario characteristics predict learners’ willingness to communicate with AI?”
- Materials and Methods
- Describe briefly the population of which the samples came from.
- Questionnaire items for AI-WTC were new, a robust instrument validation is required. Authors only used “I’m willing to…” to preserve the structure, but this does not reflect the construct as conceptualized and validated by Peng and Woodrow’s (2010) were adopted. Cronbach’s alpha is not sufficient to justify the appropriateness of the items for the constructs. More validation tests are required.
- Results and Discussion
- As mentioned above, Cronbach alpha alone is not sufficient to establish instrument reliability and validity, especially questionnaire items were newly developed. More validation tests are required.
- The use of RM-ANOVA for analysis is appropriate.
- The discussion effectively restated the main findings but lacked clarity in linking them back to the research questions.
- Conclusions
- Conclusions are appropriately presented with implications and limitations, as well as future research.
- References
- References are appropriate and up-to-date.
Author Response
Comments 1: Learner-AI interaction is not novel (lines 42-44). The claim that this interaction is novel was not supported with reference. The claim was also repeated in lines 45-46.
Response 1
We thank the reviewer for this critical observation. We agree that claiming learner-AI interaction as ‘novel’ was an overstatement that required stronger supporting references. In response, we have made the following key revisions to the introduction (see line 41-42):
- Removed the absolute term “novel” and the repetitive sentence.
- Reframed the concept to describe it as “an emerging and distinct form of social communication,” which is a more accurate and defensible characterization.
- Added a justification for study necessity by stating that this form of communication “warrants further empirical investigation,” thereby logically introducing the research gap our study aims to address.
- Strengthened the citation to include Zhang et al. (2024), which explicitly discusses the unique characteristics of AI-as-interlocutor, thereby providing scholarly support for its ‘distinct’ nature.
- Removed the repeated claim in lines 45-46.
These changes have eliminated the unsupported claim, tightened the argumentation, and strengthened the scholarly tone of the introduction. The revised sentence is as follows:
Consequently, the dialogue between a learner and an AI can be understood as an emerging and distinct form of social communication, situated within specific, task-oriented interaction scenarios and warrants further empirical investigation (Zhang et al., 2024).
Comment 2: The argument (lines 52-55) about learner acceptance and willingness to communicate in determining the value of AI appeared rather abruptly.
Response 2
We sincerely thank the reviewer for this valuable feedback. We agree that the introduction of technology acceptance and WTC as key factors needed a smoother logical transition. To address this, we have revised the second paragraph to explicitly build a causal bridge between the mixed empirical findings and our research focus. We have added a pivotal transition sentence: “The variability in these outcomes underscores that the potential of AI is not self-actualizing; it is contingent upon learners’ psychological and behavioral engagement with the technology.” This sentence directly links the “not guaranteed” effectiveness and “mixed findings” to the necessity of investigating learner-internal variables. It establishes technology acceptance and WTC not as an abrupt claim, but as the essential mediating constructs that explain why AI’s effectiveness varies, thereby providing a compelling and logical rationale for our study (see line 51-53).
Comment 3: Research question 2 starting with ‘how’ is more appropriate for qualitative study. Since this study focused solely on quantitative, quantitative research questions are critical, for example. “to what extent do scenario characteristics predict learners’ willingness to communicate with AI?”
Response 3
We thank the reviewer for the valuable feedback on formulating a more precise quantitative research question. We agree that the original “how” phrasing was more suited for qualitative inquiry. In response, we have refined Research Question 1 and 2 to better align with our repeated-measures ANOVA design, which assesses the effect of the within-subjects factor (scenario type). The revised question now reads (see page 3):
To what extent do interaction scenarios affect learners’ technology acceptance (PU and PEU)?
To what extent do interaction scenarios affect learners’ willingness to communicate with AI (AI-WTC)?
Comment 4: Describe briefly the population of which the samples came from.
Response 4
We thank the reviewer for this constructive feedback. We have revised the Participants section to provide clearer details regarding sampling and context, specifically addressing the three key points raised (see page 4):
Participant characteristics: We now explicitly report the distribution of academic backgrounds (English majors: 19.35%; other language majors: 21.53%; non-language majors: 59.13%) and academic levels (e.g., 63.49% first-year undergraduates).
Proficiency implications: The diversity in majors (particularly the inclusion of 59.13% non-language majors) is highlighted as a proxy for capturing a wide range of English proficiency levels, reflecting the heterogeneous nature of the broader EFL learner population in Chinese universities.
Prior AI exposure: We explicitly state that all participants had prior experience using generative AI tools (e.g., ChatGPT, Wenxin Yiyan) for language learning, contextualizing this within the increasing integration of such technologies in their academic environment.
These clarifications enhance the transparency and contextual grounding of the study design. The revised scripts are as follows:
The participants were 367 Chinese university students recruited from two key universities in northern China. The sample comprised both undergraduate and graduate students, with a mean age of 19.93 years. In terms of academic background, 71 students (19.35%) majored in English, 79 (21.53%) in other foreign languages, and 217 (59.13%) in non-language disciplines. Regarding academic level, 233 (63.49%) were first-year un-dergraduates, 61 (16.62%) were second-year, 33 (8.99%) were third-year, 10 (2.72%) were in their final undergraduate year, 17 (4.63%) were master’s students, and 1 participant (0.27%) was a doctoral student.
All participants had prior experience using generative AI tools (e.g., ChatGPT, Wenxin Yiyan) for auxiliary language learning purposes, as the use of such technology is increasingly integrated into the general academic environment at their institutions. The study specifically focused on the context of learning English as a foreign language and the diverse background ensured a wide range of English proficiency levels and learning needs, thereby enhancing the representativeness of the findings for the broader EFL learner population in Chinese higher education.
Comment 5: Questionnaire items for AI-WTC were new, a robust instrument validation is required. Authors only used “I’m willing to…” to preserve the structure, but this does not reflect the construct as conceptualized and validated by Peng and Woodrow’s (2010) were adopted. Cronbach’s alpha is not sufficient to justify the appropriateness of the items for the constructs. More validation tests are required.
Response 5
We thank the reviewer for this insightful comment regarding the theoretical grounding of the AI-WTC items. We agree that a robust conceptual justification is essential. The adaptation of the scale was a deliberate and theoretically driven decision (see page 4). The AI-WTC scale was adapted from Peng and Woodrow’s (2010) classroom-based WTC questionnaire, which highlights the interplay between individual and contextual variables. This foundational principle is perfectly aligned with the core design of our study, which investigates the interaction between learners’ individual dispositions (WTC and, crucially, extended to Technology Acceptance) and the contextual variable of AI interaction scenarios.
In this framework, the AI functions as “a more capable peer”. The “I am willing to…” structure was preserved to measure the well-established core of the WTC construct-behavioral intention. However, the items were necessarily re-contextualized to capture willingness to communicate with an AI partner within specific scenarios, thereby extending the original construct into the novel domain of human-AI interaction while remaining faithful to its ecological theoretical roots. The revised scripts are as follows:
The AI-WTC scale was adapted from Peng and Woodrow’s (2010) classroom-based WTC questionnaire, which highlights the interplay between individual and contextual vari-ables. Building on this foundation, the present study extended the individual dimension to include learners’ technology acceptance (PU, PEU) alongside their AI-WTC. Within a sociocultural perspective, AI was conceptualized as a “more capable peer,” with com-municative engagement understood as co-constructed through the interaction between learners’ perceptions and scenario-specific affordances. To capture this, the AI-WTC scale employed an “I am willing to …” structure across four scenarios, with six items per scenario developed in parallel with the TA scale to ensure measurement consistency.
As for the validation, we have conducted rigorous statistical analyses to address this concern, please see the detailed answer in response 6.
Commnent 6: As mentioned above, Cronbach alpha alone is not sufficient to establish instrument reliability and validity, especially questionnaire items were newly developed. More validation tests are required.
Response 6
We thank the reviewer for the critical feedback regarding the validation of our newly developed AI-WTC scale. We have thoroughly addressed this concern by performing a comprehensive set of statistical analyses, and we have revised the manuscript to include these details in the method section. Specifically, as now reported in the manuscript, we conducted a confirmatory factor analysis (CFA) using AMOS 26 (see page 7). The results confirmed that:
All standardized factor loadings were significant and exceeded 0.6, indicating strong item-construct associations.
The composite reliability (CR) for all constructs ranged from 0.897 to 0.955, demonstrating excellent internal consistency.
The average variance extracted (AVE) values ranged from 0.593 to 0.779, providing strong evidence for convergent validity.
The revised scripts are as follows:
Measurement reliability and construct validity were assessed using AMOS 26, where all standardized factor loadings exceeded 0.6 and were statistically significant (p < 0.05), indicating strong associations between each item and its respective construct and a well-fitting measurement model. Composite reliability (CR) values ranged from 0.897 to 0.955, exceeding the recommended threshold of 0.70 and demonstrating strong internal consistency. Average variance extracted (AVE) values ranged from 0.593 to 0.779, sur-passing the minimum requirement of 0.50 and supporting convergent validity. In-ter-factor correlations, calculated using SPSS 26, ranged from 0.70 to 0.91 between PU and PEU, and from 0.72 to 0.88 between TA and AI-WTC. Although correlations were rela-tively high, this pattern aligns with theoretical expectations given the close conceptual relationships among constructs. Overall, these results indicate that the instrument pro-vides a reliable and valid basis for examining technology acceptance and willingness to communicate with AI.
We believe that the combination of strong factorial validity, high reliability, satisfactory convergent validity, and theoretical coherence provides robust support for the use of our instrument in this study.
Comment 7: The discussion effectively restated the main findings but lacked clarity in linking them back to the research questions.
Response 7
We thank the reviewer for this critical observation. We agree that explicitly linking the discussion back to the research questions is essential for clarity. In direct response to this comment, we have completely restructured the Discussion section. It is now organized around directly addressing each of the three research questions in sequence, followed by a synthesis of the theoretical and practical contributions (see page 10).
The first paragraph of the discussion now explicitly addresses Research Question 1, interpreting the findings on scenario-based differences in technology acceptance (PU/PEU) and linking them to Hypothesis 1.
The second paragraph is dedicated to Research Question 2, discussing the variations in AI-WTC across scenarios and connecting them to Hypothesis 2.
The third paragraph focuses squarely on Research Question 3, analyzing the interconnections and nuanced divergences between the effects on TA and AI-WTC, and evaluating these patterns against Hypothesis 3.
This revised structure ensures a logical and transparent flow, directly demonstrating how the findings answer the specific questions posed at the outset of the study. We believe this significant revision has greatly enhanced the clarity and focus of the discussion.
Reviewer 2 Report
Comments and Suggestions for Authors
This manuscript tackles a timely question at the intersection of AI and L2 learning and offers clearly presented results with a useful comparison of interaction scenarios. The study is competently executed, and the measures and results tables are readable. That said, the framing can be more tightly anchored in theory and prior evidence: explicitly articulate the theoretical model (eg, how TAM constructs and willingness-to-communicate are expected to relate under each scenario), state hypotheses up front, and align the Discussion to those a priori expectations. Several conclusions overreach the analyses (eg, implied mediating pathways); if such mechanisms are central, conduct and report formal mediation/moderation tests with effect sizes and assumptions, or temper claims. Please clarify sampling and context (participant characteristics, proficiency, prior AI exposure), instrument validity (factor structure, reliability), and any controls. Consider potential threats (novelty, social desirability), and discuss generalisability beyond the study setting. Strengthen engagement with recent AI-in-education literature and L2 WTC research (past 3–4 years), and ensure claims about pedagogical implications track the observed effect sizes. These revisions would enhance the work.
Author Response
Comment 1: That said, the framing can be more tightly anchored in theory and prior evidence: explicitly articulate the theoretical model (eg, how TAM constructs and willingness-to-communicate are expected to relate under each scenario), state hypotheses up front, and align the Discussion to those a priori expectations.
Response 1
We sincerely thank the reviewer for this essential suggestion to strengthen the theoretical grounding and predictive rigor of our study. We fully agree that a more explicit theoretical model and a priori hypotheses are crucial for framing the research. In direct response to this comment, we have thoroughly revised the manuscript to address each point (see page3; page10):
Explicitly Articulating the Theoretical Model: We have significantly refined the theoretical framework in the introduction. Specifically, we now clearly present a model where interaction scenario characteristics are theorized to simultaneously and directly influence learners’ cognitive evaluations (Technology Acceptance: PU and PEU) and their socio-affective disposition (Willingness to Communicate with AI, AI-WTC). This model positions TA and AI-WTC as parallel outcomes, allowing us to investigate how the patterns of scenario effects on these constructs converge or diverge.
Stating Hypotheses Upfront: Based on this clarified model, we have now formulated and stated three clear, directional hypotheses before presenting the methods and results. These a priori hypotheses explicitly predict the main effects of scenarios (H1, H2) and, most importantly, the anticipated relationship between the patterns of effects on TA and AI-WTC (H3).
Aligning the Discussion to A Priori Expectations: We have carefully revised the Discussion section to structure the interpretation of findings around these pre-stated hypotheses. We now explicitly reference each hypothesis (H1, H2, H3) and discuss whether the results supported or refined our initial expectations, providing a theory-driven explanation for the observed patterns of convergence and divergence.
We believe these comprehensive revisions have tightly anchored the study within a coherent theoretical framework, greatly enhancing its conceptual clarity and scholarly contribution. We are grateful for the reviewer’s guidance in helping us achieve this.
Comment 2: Several conclusions overreach the analyses (eg, implied mediating pathways); if such mechanisms are central, conduct and report formal mediation/moderation tests with effect sizes and assumptions, or temper claims.
Response 2
We thank the reviewer for this critical observation regarding the overinterpretation of our results. We agree that claiming causal or mediating pathways without formal statistical testing was an overreach. In direct response to this comment, we have taken the following corrective actions throughout the manuscript, particularly in the Introduction, Results and Discussion sections (see page1-3; 8-11):
Removed All Implied Mediation Claims: We have eliminated all language that suggested a mediating relationship (e.g., “functions as a gateway,” “acts as a mediating scaffold”) between technology acceptance and AI-WTC. The findings are now discussed strictly in terms of observed associations and parallel patterns between the constructs across different scenarios.
Tempered Causal Language (mainly in Results section): We have replaced definitive causal language (e.g., “influences,” “shapes”) with more tentative and accurate phrasing that reflects the correlational nature of our data, such as “is associated with,” “suggests a link,” and “the patterns indicate.”
Reframed Introduction and Discussion: The Introduction and Discussions sections have been reframed to highlight the core, defensible contribution of the study: the comparative analysis of how scenario effects differentially impact TA and AI-WTC, without inferring unverified mechanisms.
By making these revisions, we have ensured that our claims are precisely aligned with the analyses we conducted.
Comment 3: Please clarify sampling and context (participant characteristics, proficiency, prior AI exposure) , instrument validity (factor structure, reliability), and any controls.
Response 3
We thank the reviewer for this constructive feedback. We have revised the Participants section to provide clearer details regarding sampling and context, specifically addressing the three key points raised:
Participant characteristics: We now explicitly report the distribution of academic backgrounds (English majors: 19.35%; other language majors: 21.53%; non-language majors: 59.13%) and academic levels (e.g., 63.49% first-year undergraduates).
Proficiency implications: The diversity in majors (particularly the inclusion of 59.13% non-language majors) is highlighted as a proxy for capturing a wide range of English proficiency levels, reflecting the heterogeneous nature of the broader EFL learner population in Chinese universities.
Prior AI exposure: We explicitly state that all participants had prior experience using generative AI tools (e.g., ChatGPT, Wenxin Yiyan) for language learning, contextualizing this within the increasing integration of such technologies in their academic environment.
These clarifications enhance the transparency and contextual grounding of the study design. The revised scripts are as follows (see page 4):
The participants were 367 Chinese university students recruited from two key universities in northern China. The sample comprised both undergraduate and graduate students, with a mean age of 19.93 years. In terms of academic background, 71 students (19.35%) majored in English, 79 (21.53%) in other foreign languages, and 217 (59.13%) in non-language disciplines. Regarding academic level, 233 (63.49%) were first-year un-dergraduates, 61 (16.62%) were second-year, 33 (8.99%) were third-year, 10 (2.72%) were in their final undergraduate year, 17 (4.63%) were master’s students, and 1 participant (0.27%) was a doctoral student.
All participants had prior experience using generative AI tools (e.g., ChatGPT, Wenxin Yiyan) for auxiliary language learning purposes, as the use of such technology is increasingly integrated into the general academic environment at their institutions. The study specifically focused on the context of learning English as a foreign language and the diverse background ensured a wide range of English proficiency levels and learning needs, thereby enhancing the representativeness of the findings for the broader EFL learner population in Chinese higher education.
Regarding validity, we have thoroughly addressed this concern by performing a comprehensive set of statistical analyses, and we have revised the manuscript to include these details in the method section. Specifically, as now reported in the manuscript, we conducted a confirmatory factor analysis (CFA) using AMOS 26 (see page7). The results confirmed that:
All standardized factor loadings were significant and exceeded 0.6, indicating strong item-construct associations.
The composite reliability (CR) for all constructs ranged from 0.897 to 0.955, demonstrating excellent internal consistency.
The average variance extracted (AVE) values ranged from 0.593 to 0.779, providing strong evidence for convergent validity.
We believe that the combination of strong factorial validity, high reliability, satisfactory convergent validity, and theoretical coherence provides robust support for the use of our instrument in this study.
Comment 4: Consider potential threats (novelty, social desirability), and discuss generalisability beyond the study setting.
Response 4
We thank the reviewer for raising the important point regarding the generalizability of our findings. We fully agree that the exclusive focus on Chinese university students is a limitation. In direct response, we have revised the Limitations section to explicitly acknowledge this. The added text concisely states that the generalizability is limited by the cultural-specific context of our sample, noting factors such as the pronounced novelty effect of AI and a potential tendency toward social desirability in the educational setting as key reasons. The specific sentence revised is: “ Second, the generalizability is limited by the Chinese student sample, as cultural-specific factors such as the pronounced novelty effect and a tendency toward social desirability in educational settings may have uniquely shaped the responses.” We believe this addition provides a clearer and more critical perspective on the boundaries of our study’s applicability. (see page 10; page 12)
Comment 5: Strengthen engagement with recent AI-in-education literature and L2 WTC research (past 3-4 years)
We thank the reviewer for the valuable suggestion to strengthen our engagement with recent literature. We have thoroughly revised the manuscript to integrate key contemporary studies from the past 3-4 years, particularly within the literature review and theoretical framework sections. Specifically, in the revised introduction to WTC (as shown in the provided excerpt), we have:
Cited a seminal 2021 systematic review on WTC (Nematizadeh, 2021) to establish its dynamic and situation-specific nature, reflecting the most current theoretical understanding.
Incorporated very recent empirical studies from 2024 (Fathi et al., 2024; Wang et al., 2024) that directly investigate WTC in AI contexts. These references are crucial as they not only validate the application of WTC to AI environments but also help define the unique characteristics of the AI-WTC construct, precisely addressing the reviewer's point.
These additions, along with other recent works integrated throughout the paper, ensure that our study is firmly situated within the ongoing scholarly conversation on AI in education and L2 WTC, providing a contemporary and robust theoretical foundation. The revised scripts are as follows (page 2):
Beyond technology adoption, learners’ communicative engagement with AI also hinges on affective and psychological readiness, as explained by Willingness to Com-municate (WTC) theory. WTC describes an individual’s readiness to engage in discourse (McCroskey & Baer, 1985; MacIntyre et al., 1998) and has been widely applied in second language acquisition research. In the L2 context, WTC is recognized not merely as a stable trait but as a dynamic, situation-specific construct that is susceptible to a complex interplay of linguistic, affective, and situational variables (Nematizadeh; 2021). With the rise of digital technologies, L2 WTC research expanded into computer-mediated communication. Early findings not only validate the applicability of the WTC framework to AI contexts but are also refining its contours, revealing critical distinctions while AI can offer a low-anxiety space, learners’ willingness to engage is profoundly shaped by their perception of the AI’s competence, reliability, and the specific purpose of the interaction (Fathi et al., 2024; Wang et al., 2024). This evolving line of inquiry not only demonstrates the adaptability of the WTC framework to AI-mediated ecologies but also highlights that AI-WTC is a unique construct sensitive to technological and contextual affordances.
Comment 6: and ensure claims about pedagogical implications track the observed effect sizes.
Response 6
We thank the reviewer for this critical reminder to maintain precise alignment between our empirical findings and the practical claims derived from them. In direct response, we have carefully reviewed and refined the language throughout the ‘Pedagogical Implications’ section to ensure that our recommendations accurately reflect the modest yet significant effect sizes observed (see page 12). Specifically, we have replaced definitive or overly strong claims with more calibrated language that acknowledges the nuanced role of scenario design. For instance, we now state that sequencing tasks from high to low AI-compatibility “may be a prudent pedagogical strategy” and that the observed effects “nonetheless indicate” its potential value. This phrasing showcased that scenario type is one meaningful factor among others influencing learner perceptions, preventing overstatement of its impact while still advocating for its considered application in instructional design.
Reviewer 3 Report
Comments and Suggestions for Authors
The article is very interesting and deals with the highly relevant question of students' willingness to communicate with AI in English. However, a few corrections and clarifications would certainly improve its quality:
- First of all, the Authors do not state clearly enough that the study concerns the use of AI in learning English and not any language. This can be inferred from certain statements (e.g. on page 6 the Authors mention 'looking up the meanings of unfamiliar English words' and English appears many times in the questionnaire, e.g. 'It would be easy for me to become skillful at using AI to understand English texts
and their structure'), but at the beginning the reader might assume the study concerns foreign language communication in general. The fact that the study focuses on learning English should be stated in the abstract and in the introduction in order to make it clear to the reader. - On page 2, the Authors say: 'few studies have concurrently measured PU, PEU, and AI-WTC within the same quantitative framework.' One might actually wonder what they mean by 'the same quantitative framework': whether it is the same set of statistical tests and, if so, for perceived usefulness, perceived ease of use and willingness to communicate, or for the studies (i.e. whether Fathi et al. 2023 and Zhang et al. 2024 used the same statistical tests as the present Authors, etc.)
- Page 4: 'To validate and refine this a priori framework, we administered an open-ended questionnaire to 67 university-level EFL learners (...).' The reader might wonder whether these are the same students as those who completed the final questionnaire, or different ones. I would suggest stating this clearly.
- Page 4: 'Using a three-phase thematic coding procedure (...).' It might be advisable to describe these three phases briefly.
- Page 5: The students evaluated 24 interaction tasks, but this begs the question of whether they were really familiar with all the task types, or maybe not necessarily. Maybe there were students who had never done a specific task type and still evaluated it. It should be stated whether they were really familiar with the tasks.
-
Page 11. The Authors write: 'Factual information retrieval consistently received the highest ratings for PU, PEU, and WTC, suggesting that learners perceive this type of interaction as highly beneficial, accessible, and conducive to communication.' Possibly, this is what AI seems to be good at and useful for. Maybe this kind of communication is not necessarily perceived as beneficial, but, first of all, accessible, given the characteristics of AI. However, this is only a suggestion for the Authors to consider.
-
Page 11: 'The following sections delve deeper into this dynamic, first examining how facilitating scenarios (factual information retrieval scenario & academic knowledge inquiry scenario).' One might wonder why they are called 'facilitating,' as, in fact, for example, advice of foreign language learning may also be facilitating, though in a different way. Perhaps such scenarions (factual information retrieval scenario & academic knowledge inquiry scenario) are simply more compatible with AI and its use.
-
Page 12: The Authors write: 'AI fulfills its role as a More Capable Other.' In the rest of the article, they use the term 'more capable peer' and, if this is Vygotsky's term, it should be used consistently.
-
Page 12: The Authors use the abbreviation AIS as if it were obvious to readers: 'However, the observed divergences, particularly the dissociation between AIS’s moderate perceived usefulness ratings.' Even though it is explained in the abbreviations (page 14) as an advisory interaction scenario, the reader might feel confused and not necessarily look up the abbreviation. I would suggest adding an explanation of 'AIS' on page 12.
- Page 12: 'For AI to function as a “more capable peer,” designers must prioritize adaptive scaffolding that reduces learner-side negotiation costs, while educators should train learners to strategically navigate AI limitations,
a dual approach that strengthens technology acceptance and fosters greater willingness to communicate with AI (Zhang & Zhang, 2024).' Maybe AI was not originally designed to be an English teacher and maybe it was assumed that people would use their native language in the advisory interaction scenario? Maybe this should be stated more explicitly.
11. Even though the Authors mention that the questionnaire consisted of a Likert scale, the questionnaire in Appendix 1 contains only the statements and a reader who no longer remembers the description of the method might think these were yes/no questions. I suggest adding numbers from 1 to 5 to the questionnaire or at least beginning the questionnaire with the information that it was a 1 to 5 Likert scale.
Comments on the Quality of English Language
As for the Authors' English, it is generally very good, but a few mistakes need correcting:
1. Page 1: 'pedagogically guidance' should be changed to 'pedagogical guidance.'
2. Page 2: It seems strange that the Authors call AI 'an opinion or advice giver.' Why not an advisor, as in the rest of the article?
3. The abbreviation 'e.g.' is written with two full stops: after e and after g, while the Authors write 'eg.' I recommend changing it to 'e.g.'
4. Page 8: 'To examine whether learners’ technology acceptance, operationalized as Perceived Usefulness (PU) and Perceived Ease of Use (PEU), varied significantly across the four AI interaction scenarios. A within-subjects design was employed, as the same group of participants provided ratings for each construct under all four scenario conditions: (...)' The first sentence is unfinished, so both sentence parts should be combined into one sentence: 'To examine whether learners’ technology acceptance, operationalized as Perceived Usefulness (PU) and Perceived Ease of Use (PEU), varied significantly across the four AI interaction scenarios, a within-subjects design was employed, as the same group of participants provided ratings for each construct under all four scenario conditions: (...).'
5. In the conclusions, the Authors tend to repeat the verb 'underscore': 'These findings collectively underscore the central role of context', 'The findings underscore that effective AI integration in language education requires intentional scenario design.' Maybe they should change the verb, e.g. 'The findings demonstrate that effective AI integration in language education requires intentional scenario design.'
6. Page 11: A sentence begins with 'whereas their work established AI‘s general legitimacy as a communicative partner.' 'Whereas' at the beginning of the sentence should be capitalized.
7. On page 14, there is a typo: 'Perceived Ease of Us.' It should be: 'Perceived Ease of Use.'
8. Page 15: 'Communicating with AI to discuss research directions or ideas would enhance my effectiveness on academic writing.' Better: 'Communicating with AI to discuss research directions or ideas would enhance my effectiveness in academic writing.'
Author Response
Comment 1: First of all, the Authors do not state clearly enough that the study concerns the use of AI in learning English and not any language. This can be inferred from certain statements (e.g. on page 6 the Authors mention ‘looking up the meanings of unfamiliar English words’ and English appears many times in the questionnaire, e.g. ‘It would be easy for me to become skillful at using AI to understand English texts and their structure’), but at the beginning the reader might assume the study concerns foreign language communication in general. The fact that the study focuses on learning English should be stated in the abstract and in the introduction in order to make it clear to the reader.
Response 1
We sincerely thank the reviewer for this critical observation. We agree that the focus on English learning should be explicitly stated upfront to avoid any ambiguity. In direct response to this comment, we have made the following revisions to ensure clarity:
In the Title: The Effects of Interaction Scenarios on EFL Learners’ Technology Acceptance and Willingness to Communicate with AI (see page 1);
In the Abstract: Drawing on a sociocultural perspective, this study examines how different AI interaction scenarios influence Chinese EFL learners’ technology acceptance (TA), constructed as perceived usefulness (PU) and perceived ease of use (PEU), and their willingness to communicate with AI (AI-WTC). This immediately establishes the context of English as a Foreign Language for the reader (see page 1).
In the Introduction: We have added an explicit statement in the conclusion paragraphs that Drawing on sociocultural theory, this study focuses on Chinese EFL learners, examining how scenarios shape their technology acceptance and willingness to communicate with AI. ensuring the scope is clear from the outset. (see page 3)
In the Participants and procedure part, we have now clearly stated: The study specifically focused on the context of learning English as a foreign language and the diverse background ensured a wide range of English proficiency levels and learning needs, thereby enhancing the representativeness of the findings for the broader EFL learner population in Chinese higher education. (see page 4)
These changes ensure that the study’s focus on the English language is unambiguously communicated to the reader at the very beginning of the manuscript and consistently throughout.
Comment 2: On page 2, the Authors say: ‘few studies have concurrently measured PU, PEU, and AI-WTC within the same quantitative framework.’ One might actually wonder what they mean by ‘the same quantitative framework’: whether it is the same set of statistical tests and, if so, for perceived usefulness, perceived ease of use and willingness to communicate, or for the studies (i.e. whether Fathi et al. 2023 and Zhang et al. 2024 used the same statistical tests as the present Authors, etc.)
Response 2
Thank you for this helpful comment. We realize that our earlier wording could cause ambiguity, as “quantitative framework” might be interpreted as referring to specific statistical methods. What we intended to convey is that while TA (PU and PEU) and AI-WTC have often been investigated separately, few studies have integrated these constructs into a unified analytical framework at the construct level, particularly in AI-mediated learning contexts. To clarify, we have revised the text to emphasize the need for a scenario-sensitive analytical framework that captures the variability of AI interaction scenarios and their joint influence on PU, PEU, and AI-WTC. The revised scripts are as follows (see page 2):
While TA and AI-WTC have often been examined separately, their integration within AI-mediated learning contexts remains scarce. This gap is compounded by a tendency in prior research to treat AI as a monolithic entity, thereby overlooking the fundamental ways in which specific interaction scenarios shape learner perceptions and behaviors. This limitation points to the need for a more differentiated analytical framework that can capture the variability of AI interaction scenarios, a perspective reinforced by emerging studies showing that learners engage with AI in distinct, role-based capacities (Peng & Liang, 2025). In this light, a scenario-sensitive perspective offers a promising direction for understanding how perceived usefulness (PU), perceived ease of use (PEU), and willingness to communicate with AI (AI-WTC) interact in more nuanced ways.
Comment 3: Page 4: ‘To validate and refine this a priori framework, we administered an open-ended questionnaire to 67 university-level EFL learners (...).’ The reader might wonder whether these are the same students as those who completed the final questionnaire, or different ones. I would suggest stating this clearly.
Response 3
We thank the reviewer for this important request for clarification. The 67 participants in the preliminary qualitative phase constituted a separate, independent sample. They were recruited from the same two universities that defined the target population for the main study, but they were not the same individuals who later completed the final quantitative questionnaire. This approach of using a separate sample for item development helps to ensure that the scenarios were grounded in authentic learner experiences before being evaluated by a new cohort of participants, thereby enhancing the ecological validity of the instrument. We revised the scripts as follows(see page 5):
To develop ecologically valid scenarios for the subsequent scale, we first administered an open-ended questionnaire to a separate, independent sample of 67 university-level EFL learners, eliciting detailed accounts of their actual and envisioned AI-supported learning activities.
Comment 4: Page 4: ‘Using a three-phase thematic coding procedure (...).’ It might be advisable to describe these three phases briefly.
Response 4
We thank the reviewer for this helpful suggestion. In response, we have revised this section to include a concise description of the three-phase thematic analysis procedure. The updated text now explicitly outlines the stages of open coding, thematic categorization, and scenario refinement to enhance methodological transparency while maintaining brevity, see below (see page 5):
To develop ecologically valid scenarios for the subsequent scale, we first administered an open-ended questionnaire to a separate, independent sample of 67 university-level EFL learners, eliciting detailed accounts of their actual and envisioned AI-supported learning activities (938 words). The qualitative data underwent a three-phase thematic analysis involving initial open coding to identify instances of AI use, categorization of codes into broader themes, and consolidation into four overarching scenarios. This process yielded 130 concrete interaction instances and six representative tasks were then selected per scenario based on frequency and pedagogical salience, resulting 24 balanced and reality-based tasks for the subsequent scales (DeVellis & Thorpe, 2021).
Comment 5: Page 5: The students evaluated 24 interaction tasks, but this begs the question of whether they were really familiar with all the task types, or maybe not necessarily. Maybe there were students who had never done a specific task type and still evaluated it. It should be stated whether they were really familiar with the tasks.
Response 5
We thank the reviewer for raising this important point regarding task familiarity. We agree that this is a critical aspect for the validity of the responses.
The 24 interaction tasks were directly derived from empirical data, specifically, an open-ended questionnaire administered to a separate sample of 67 university-level EFL learners from the target population. Participants in that preliminary study described their actual and envisioned use of AI for language learning. Thematic analysis of their responses yielded a diverse set of behaviors, from which we selected the most frequently reported and pedagogically significant tasks to form the 24 scenarios. Therefore, these tasks are ecologically valid as they represent authentic interactions within the lived experience of the target learner population. While it is possible that any single participant may not have engaged with every single task type, the tasks as a whole reflect a collectively familiar repertoire of AI uses. This approach ensures that the scenarios are contextually grounded and meaningful for the participants, strengthening the validity of their evaluations.
The revised scripts are as follows (page 5):
To develop ecologically valid scenarios for the subsequent scale, we first administered an open-ended questionnaire to a separate, independent sample of 67 university-level EFL learners, eliciting detailed accounts of their actual and envisioned AI-supported learning activities (938 words). The qualitative data underwent a three-phase thematic analysis involving initial open coding to identify instances of AI use, categorization of codes into broader themes, and consolidation into four overarching scenarios. This process yielded 130 concrete interaction instances. Six representative tasks were selected per scenario based on frequency and pedagogical salience (DeVellis & Thorpe, 2021), with each task either reflecting experiences frequently reported in the open-ended responses or clearly described with sufficient context to ensure participants’ familiarity, so that students could meaningfully evaluate them even if they had not previously performed a specific task.
Comment 6: Page 11. The Authors write: ‘Factual information retrieval consistently received the highest ratings for PU, PEU, and WTC, suggesting that learners perceive this type of interaction as highly beneficial, accessible, and conducive to communication.’ Possibly, this is what AI seems to be good at and useful for. Maybe this kind of communication is not necessarily perceived as beneficial, but, first of all, accessible, given the characteristics of AI. However, this is only a suggestion for the Authors to consider.
Response 6
We thank the reviewer for this insightful observation regarding the interpretation of why the Factual Information Retrieval Scenario (FIRS) received the highest ratings. This is a valuable distinction. We have fully incorporated this suggestion into the revised Discussion section (see page 10).
Specifically, we have refined our interpretation to emphasize that the primary strength of FIRS lies in its accessibility and high compatibility with AI’s core capabilities, which leads to low cognitive load for learners. We have tempered the claim of inherent “benefit” and instead focus on the scenario’s structural alignment with what AI currently does best, making the interaction feel effortless and predictable. This shift in phrasing provides a more precise and theoretically grounded explanation.
Comment 7: Page 11: ‘The following sections delve deeper into this dynamic, first examining how facilitating scenarios (factual information retrieval scenario & academic knowledge inquiry scenario).’ One might wonder why they are called ‘facilitating,’ as, in fact, for example, advice of foreign language learning may also be facilitating, though in a different way. Perhaps such scenarios (factual information retrieval scenario & academic knowledge inquiry scenario) are simply more compatible with AI and its use.
Response 7
We thank the reviewer for this insightful suggestion regarding the term “facilitating scenarios.” We agree that the descriptor “facilitating” could be subjective and might not fully capture the underlying reason for the observed effects. In response, we have removed the term “facilitating” from the revised Discussion section. Instead, we now frame the analysis around the core concept of “scenario-AI compatibility,” focusing on the alignment between the specific demands of a scenario and the current functional strengths and accessibility of AI technology. This shift in terminology provides a more precise, objective, and mechanism-based explanation for why scenarios like factual information retrieval are consistently rated more positively, as it directly links the findings to the affordances of the technology itself (see page 10).
Comment 8: Page 12: The Authors write: ‘AI fulfills its role as a More Capable Other.’ In the rest of the article, they use the term ‘more capable peer’ and, if this is Vygotsky’s term, it should be used consistently.
Response 8
Thank you for pointing out the inconsistency. We have revised the manuscript to use “more capable peer” consistently throughout the text. The term has been updated on page 12 and all other instances to ensure terminological uniformity (see page 12).
Comment 9: Page 12: The Authors use the abbreviation AIS as if it were obvious to readers: ‘However, the observed divergences, particularly the dissociation between AIS’s moderate perceived usefulness ratings.’ Even though it is explained in the abbreviations (page 14) as an advisory interaction scenario, the reader might feel confused and not necessarily look up the abbreviation. I would suggest adding an explanation of ‘AIS’ on page 12.
Response 9
We thank the reviewer for this thoughtful suggestion aimed at improving the reader-friendliness of the manuscript. We agree that minimizing any potential confusion for the reader is of utmost importance. In response to this comment, we have ensured that upon their every appearance in the Results section and Discussion section (and indeed throughout the manuscript), all scenario names including Advisory Interaction Scenario (AIS), Language Skills Support Scenario (LSSS), Academic Knowledge Inquiry Scenario (AKIS), and Factual Information Retrieval Scenario (FIRS), are now explicitly presented with both their full name and abbreviation.
This revision provides immediate clarity within the main text, eliminating the need for the reader to refer back to the abbreviations list and thereby enhancing the reading experience.
Comment 10: Page 12: ‘For AI to function as a “more capable peer,” designers must prioritize adaptive scaffolding that reduces learner-side negotiation costs, while educators should train learners to strategically navigate AI limitations, a dual approach that strengthens technology acceptance and fosters greater willingness to communicate with AI (Zhang & Zhang, 2024).’ Maybe AI was not originally designed to be an English teacher and maybe it was assumed that people would use their native language in the advisory interaction scenario? Maybe this should be stated more explicitly.
Response 10
We sincerely thank the reviewer for raising this profoundly insightful point regarding the inherent design and assumed linguistic context of AI interactions. We fully agree that explicitly acknowledging AI’s original purpose as a general-purpose tool, not a dedicated language teacher, is critical to framing its pedagogical role realistically.
In direct response to this comment, we have moved beyond a simple clarification and have integrated the core insight as a foundational element of the ‘Pedagogical Implications’ section. Specifically, we now explicitly state that “most AI systems were not originally designed as language teachers but rather as general-purpose conversational agents,” and that expectations should be recalibrated by positioning AI as a scaffolding tool or “more capable peer.” This reframing directly addresses the reviewer’s point by emphasizing that AI’s core competencies lie in information processing, not in replicating the nuanced guidance of a human educator.
Furthermore, we have incorporated the reviewer’s astute suggestion regarding native language use as a strategic pedagogical scaffold. The revised text now proposes that “for highly complex scenarios like Advisory Interaction (AIS), strategically permitting the use of the native language can serve as a transitional scaffold… allowing learners to focus on complex advisory tasks before transitioning to the target language.” This acknowledges the cognitive reality of L1 use in complex reasoning and aligns with translanguaging pedagogy (see page 12).
Comment 11: Even though the Authors mention that the questionnaire consisted of a Likert scale, the questionnaire in Appendix 1 contains only the statements and a reader who no longer remembers the description of the method might think these were yes/no questions. I suggest adding numbers from 1 to 5 to the questionnaire or at least beginning the questionnaire with the information that it was a 1 to 5 Likert scale.
Response 11
We thank the reviewer for this helpful suggestion. To clarify the response format, we have now added the following statement at the beginning of Appendix 1: “All items were measured on a 5-point Likert scale, ranging from 1 (strongly disagree) to 5 (strongly agree).” This revision ensures that readers can easily understand the scale used for all questionnaire items (see page14).
Comment 12: Comments on the Quality of English Language
- Page 1: ‘pedagogically guidance’ should be changed to ‘pedagogical guidance.’
- Page 2: It seems strange that the Authors call AI ‘an opinion or advice giver.’ Why not an advisor, as in the rest of the article?
- The abbreviation ‘e.g.’ is written with two full stops: after e and after g, while the Authors write ‘eg.’ I recommend changing it to ‘e.g.’
- Page 8: ‘To examine whether learners’ technology acceptance, operationalized as Perceived Usefulness (PU) and Perceived Ease of Use (PEU), varied significantly across the four AI interaction scenarios. A within-subjects design was employed, as the same group of participants provided ratings for each construct under all four scenario conditions: (...)’ The first sentence is unfinished, so both sentence parts should be combined into one sentence: ‘To examine whether learners’ technology acceptance, operationalized as Perceived Usefulness (PU) and Perceived Ease of Use (PEU), varied significantly across the four AI interaction scenarios, a within-subjects design was employed, as the same group of participants provided ratings for each construct under all four scenario conditions: (...).’
- In the conclusions, the Authors tend to repeat the verb ‘underscore’: ‘These findings collectively underscore the central role of context’, ‘The findings underscore that effective AI integration in language education requires intentional scenario design.’ Maybe they should change the verb, e.g. ‘The findings demonstrate that effective AI integration in language education requires intentional scenario design.’
- Page 11: A sentence begins with ‘whereas their work established AI’s general legitimacy as a communicative partner.’ ‘Whereas’ at the beginning of the sentence should be capitalized.
- On page 14, there is a typo: ‘Perceived Ease of Us.’ It should be: ‘Perceived Ease of Use.’
- Page 15: ‘Communicating with AI to discuss research directions or ideas would enhance my effectiveness on academic writing.’ Better: ‘Communicating with AI to discuss research directions or ideas would enhance my effectiveness in academic writing.
Response 12
We sincerely thank the reviewer for their meticulous review and valuable suggestions regarding the quality of English language and formatting throughout the manuscript. We have carefully addressed all the points raised, including:
Correcting grammatical errors (e.g., “pedagogically guidance” to “pedagogical guidance”).
Improving sentence structure and fluency (e.g., combining the fragmented sentence on Page 8).
Ensuring consistent and appropriate terminology (e.g., using “advisor” throughout).
Correcting typographical and formatting errors (e.g., “e.g.”, “Whereas”, “Perceived Ease of Us”).
Enhancing word choice variety (e.g., replacing the repetitive use of “underscore” in the conclusions).
Furthermore, as indicated in our responses to other comments, the extensive revisions made to the Discussion and Conclusion sections provided an opportunity to comprehensively refine the language and style in those parts of the manuscript.
Reviewer 4 Report
Comments and Suggestions for Authors
See attached.

Can be improved because part of the work sounds like AI writing.
Author Response
Comment 1: Use of self-reporting: Relying solely on surveys is limiting regarding what learners actually do. The validity would be increased by triangulation with behavioural or qualitative data.
Response 1
We thank the reviewer for this insightful suggestion. We completely agree that triangulation with behavioral data would provide a deeper layer of understanding. We have explicitly acknowledged this as an important limitation in the revised manuscript. The primary reason for not collecting behavioral data in this phase was rooted in methodological design and ethical considerations. To ensure high response rates and sample representativeness for our large-scale (N=367), scenario-comparison study, we prioritized strict participant anonymity. While this approach successfully captured broad perceptual patterns, it inherently precluded the possibility of post-hoc behavioral tracking, as no identifying information was collected that would allow for follow-up data collection. We have therefore framed this limitation as a key direction for future research, suggesting that a subsequent mixed-methods study with a smaller, consenting cohort could richly complement our findings with detailed behavioral analytics (see page12). The revised scripts are as follows
First, the reliance on self-reported data without behavioral triangulation may not fully capture actual engagement, as responses could be influenced by social desirability bias.
Comment 2: Sample limitations: A university sample only in China is limiting for generalizability. The cultural factor deserves to be taken more into account.
Response 2
We thank the reviewer for rightly pointing out the limitations regarding the generalizability of our findings, a point we take very seriously.
We have addressed this concern in two key ways (see page 10-12):
Integrating Cultural Context into Interpretation: More importantly, following the reviewer’s advice to take cultural factors into account, we have moved beyond a simple acknowledgment. We have actively integrated consideration of the specific Chinese EFL learning context into our socioculturally-grounded discussion. Specifically, we discuss how educational traditions that emphasize structured tasks and clear role definitions might shape learners’ perceptions of ease of use and willingness to communicate in different scenarios (e.g., making open-ended advisory interactions particularly challenging). Thus, rather than treating the cultural specificity solely as a limitation, we have leveraged it to provide a deeper, more nuanced interpretation of our findings.
Explicit Acknowledgment in Limitations: As the reviewer suggested, we have explicitly acknowledged in the ‘Limitations’ section that the exclusive use of a Chinese university sample restricts the generalizability of the findings to other cultural and educational contexts.
We believe this approach strengthens the manuscript by transparently addressing the sampling boundary while using the cultural context to generate richer theoretical insight.
Comment 3: Interpretation of effect size: Reported η2 are small to moderate but interpretation is sometimes misleading with regard to the practical meaning.
Response 3
We sincerely thank the reviewer for raising this crucial point. We acknowledge that our initial drafts sometimes used language that overstated the practical meaning of the small-to-moderate η² values. We have thoroughly revised the manuscript to ensure that our claims are logically matched to the proportion of variance accounted for and to prevent any potential misinterpretation. The key revisions, implemented throughout the Results and Discussion sections (see page 8-11), include:
Explicitly Stating Variance Explained: Following each report of a significant effect, we now immediately clarify the practical meaning of the η² value. For instance, we state that a result explains “a small proportion of the variance” or “a limited portion,” ensuring the reader understands the scale of the effect from the outset.
Adopting Tentative Language: We have replaced strong, definitive verbs (e.g., “delineated,” “demonstrated”) with more cautious and accurate language (e.g., “suggested a pattern,” “indicated,” “highlighted differential sensitivity”). This shift better reflects the nuanced nature of findings based on effect sizes that, while statistically significant, account for a limited portion of the variance.
We believe these comprehensive changes have significantly enhanced the methodological rigor and interpretive accuracy of our manuscript. Our claims are now carefully calibrated to the strength of the empirical evidence, and we are grateful for the reviewers’ feedback in helping us achieve this.
Comment 4: Redundancy and length: Theoretical framing (in particular, ZPD and “more capable peer”) is repeated too much, making it less concise.
Response 4
We appreciate the reviewer’s valuable comment on the redundancy of the theoretical framing, particularly regarding the Zone of Proximal Development (ZPD) and the notion of the “more capable peer.” In the revised manuscript, we have streamlined the theoretical framing in the Introduction (see page 3) by reducing overlapping explanations and integrating the key points more concisely. Similarly, in the Discussion section (see page10-11), we have simplified references to these concepts, ensuring that they support the interpretation of findings without unnecessary repetition. We believe these revisions have improved the clarity and conciseness of the manuscript.
Comment 5: Reporting problems: A number of ANOVA results use incomplete degrees of freedom (e.g., “F(3, df)”). This ought to be adjusted to achieve transparency.
Response 5
We sincerely thank the reviewer for pointing out this oversight in our statistical reporting. We have carefully reviewed and corrected all ANOVA results throughout the manuscript to ensure full transparency. The degrees of freedom have been completed based on the analysis design, and all the results have been updated accordingly in the Results section (see page 8).
The results, presented in Table 2, revealed a statistically significant main effect of interaction scenario on PU, F(3, 1098) = 16.096, p < .001, η² = .042, and on PEU, F(3, 1098) = 24.641, p < .001, η² = .063. As shown in Table 3, the analysis identified a significant main effect of interaction scenario on AI-WTC, F(3, 1098) = 9.600, p < .001, η² = .026.
Comment 6: Terminology inconsistency In a few parts “Perceived Usefulness (PEU)” is misapplied while actual one was “Perceived Ease of Use (PEU)”.
Response 6
We sincerely thank the reviewer for their meticulous attention to detail in identifying this terminology error. The conflation of ‘Perceived Usefulness’ with the ‘PEU’ acronym was indeed incorrect. We have carefully reviewed the entire manuscript and corrected this inconsistency. The primary occurrence was in the questionnaire items within Appendix I (see page14-17), where the labels have now been standardized to accurately reflect ‘Perceived Ease of Use (PEU)’.
Comment 7: References: Heavily focused on recent 2024–2025 works; more conversation with previous AI-in-education work is necessary.
Response 7
We appreciate the reviewer’s observation regarding the temporal balance of the literature review. Indeed, our initial focus was heavily on works from 2024–2025, reflecting the surge of AI-in-education research in recent years. In the revised manuscript, we have supplemented the discussion with earlier studies, including two works published in 2023, to provide a more balanced coverage. Furthermore, while much of the literature is recent, our study is theoretically anchored in the well-established Technology Acceptance Model (Davis, 1989), a classic framework that we have deliberately applied to contemporary AI-mediated learning contexts. By integrating both earlier foundational studies and the most up-to-date research, we believe the literature review now presents a clearer and more coherent trajectory of how TAM and WTC frameworks have been extended into the domain of AI in education (see page1-3).
AlAfnan, M., Dishari, S., Jovic, M., & Lomidze, K. (2023). ChatGPT as an educational tool: Opportunities, challenges, and recommendations for communication, business writing, and composition courses. Journal of Artificial Intelligence and Technology. https://doi.org/10.37965/jait.2023.0184
Fauzi, F., Tuhuteru, L., Sampe, F., Ausat, A. M. A., & Hatta, H. R. (2023). Analysing the role of CHAT-GPT in improving student productivity in higher education. Journal on Education, 5(4), 14886– 14891. https://doi.org/10.31004/joe.v5i4.2563
Comment 8: Style: There are a few grammatical errors here and there, and the manuscript can further be polished for better flow and brevity.
Response 8
We sincerely thank the reviewer for this valuable feedback on improving the manuscript’s clarity and style. We have thoroughly proofread the entire text to correct grammatical errors and refine the language for better flow. Specifically, we have streamlined several sections for brevity, such as condensing the description of the 24-scenario development process in the questionnaire design section. These revisions have enhanced the overall readability and conciseness of the manuscript without compromising the methodological details.
Comment 9: ChatGPT acknowledgement: This would be transparent but may be editorially problematic. Make sure contributors adhere to editorial polices and that authors assume full responsibility for the manuscript.
Response 9
We thank the reviewer for raising this important point regarding the use of AI tools. We confirm that we have strictly adhered to the journal’s editorial policies. In accordance with these policies, we have acknowledged the use of AI tools for language polishing and refinement in the Acknowledgements section.
Comment 10: Explanation of contribution: Describe how this research expands current applications of TAM/WTC in AI-based settings (e.g., online Recommendation Agents).
Response 10
We thank the reviewer for the opportunity to clarify our study’s contributions to expanding TAM and WTC within AI-based settings. In direct response to this comment, we have revised the Discussion section to include a dedicated paragraph (the last paragraph) that explicitly outlines these contributions. Our research provides key advancements on both theoretical and methodological fronts (page 11):
Theoretical Contribution: Our primary contribution is the introduction of a scenario-sensitive perspective into the application of TAM and WTC. We move beyond evaluating AI as a monolithic tool to demonstrate that the relationship between technology acceptance and willingness to communicate is not static, but dynamically contingent on the specific interaction scenario.
Methodological Contribution: To operationalize this perspective, the study develops and validates a novel scenario-based instrument capable of capturing nuanced variations in user perceptions across different AI-mediated tasks. This methodological innovation provides a reliable tool for future research to continue investigating the critical role of context, moving the field toward more differentiated analyses.
Comment 11: Streamline the conversation: Lessen minimizes the repetitive sociocultural framing of the discussion, while highlighting distinctive findings from the data.
Response 11
We thank the reviewer for the valuable suggestion to streamline the discussion and more prominently feature our distinctive empirical findings. We agree that a focused presentation enhances the impact of the study. In direct response to this comment, we have carefully revised both the Introduction and the Discussion sections.
Introduction: We have condensed the preliminary theoretical discussion on sociocultural theory, presenting it in a more concise manner to set the stage without overemphasis (page 3).
Discussion: We have significantly restructured this section to reduce repetitive sociocultural framing. The revised discussion now leads with the distinctive patterns observed in the data, such as the differential sensitivity of PEU versus PU and AI-WTC to scenario complexity, and the critical divergence between perceived utility and communicative willingness in advisory scenarios. Sociocultural theory is now invoked more selectively to interpret these specific findings, rather than serving as a repetitive overarching frame (pages 10-11).
Comment 12: Interpret effect sizes prudently: Match claims more logically with small proportion of variance accounted for.
Response 12
We sincerely thank the reviewer for raising this crucial point. We acknowledge that our initial drafts sometimes used language that overstated the practical meaning of the small-to-moderate η² values. We have thoroughly revised the manuscript to ensure that our claims are logically matched to the proportion of variance accounted for and to prevent any potential misinterpretation. The key revisions, implemented throughout the Results and Discussion sections (pages 8-11), include:
Explicitly Stating Variance Explained: Following each report of a significant effect, we now immediately clarify the practical meaning of the η² value. For instance, we state that a result explains “a small proportion of the variance” or “a limited portion,” ensuring the reader understands the scale of the effect from the outset.
Adopting Tentative Language: We have replaced strong, definitive verbs (e.g., “delineated,” “demonstrated”) with more cautious and accurate language (e.g., “suggested a pattern,” “indicated,” “highlighted differential sensitivity”). This shift better reflects the nuanced nature of findings based on effect sizes that, while statistically significant, account for a limited portion of the variance.
We believe these comprehensive changes have significantly enhanced the methodological rigor and interpretive accuracy of our manuscript. Our claims are now carefully calibrated to the strength of the empirical evidence, and we are grateful for the reviewers’ feedback in helping us achieve this.
Comment 13: Enhance practical implications: Provide examples on how educators can create valuable AI- mediated tasks (e.g., the scaffolding prompts, the hybrid human-AI support).
Response 13
We thank the reviewer for this constructive suggestion to enhance the practical value of our findings. We agree that concrete examples are essential for bridging research and practice. In direct response, we have significantly enriched the ‘Pedagogical Implications’ section (page 12) by incorporating precise, actionable examples of AI-mediated task design as suggested.
Scaffolding Prompts: We now provide a specific example of a curated prompt template for the complex Advisory Interaction Scenario (AIS): “Act as a critical friend and evaluate the thesis statement of my essay...” This moves beyond a vague instruction and demonstrates how to structure a prompt for productive AI interaction.
Hybrid Human-AI Support: We explicitly outline a model for hybrid support by suggesting that educators employ “AI for initial support while reserving human oversight for higher-order tasks such as assessing argument coherence and cultural appropriateness.” This provides a clear framework for integrating AI within a broader pedagogical process.
We believe these concrete illustrations transform our pedagogical implications into a more valuable and directly applicable guide for educators.
Comment 14: Widen literature: Draw on a variety of sources, especially Western and comparative studies of AI in language education.
Response 14
We sincerely thank the reviewer for this valuable suggestion to broaden the scholarly perspective of our manuscript. In direct response to this comment, we have significantly widened the scope of our literature review by incorporating several key international and comparative studies.
Specifically, we have now integrated the following works into the introduction and theoretical framework sections: (page 1-3)
Apriani, E., Cardoso, L., Obaid, A. J., Muthmainnah, Wijayanti, E., Esmianti, F., & Supardan, D. (2024). Impact of AI-powered chatbots on EFL students’ writing skills, self-efficacy, and self-regulation: A mixed-methods study. Global Educational Research Review, 1(2), 57–72. https://doi.org/10.71380/GERR-08-2024-8
KarataÅŸ, F., Abedi, F. Y., Gunyel, F. O., Karadeniz, D., & Kuzgun, Y. (2024). Incorporating AI in foreign language education: An investigation into ChatGPT’s effect on foreign language learners. Education and Information Technologies, 29, 19343–19366. https://doi.org/10.1007/s10639-024-12574-6
Chiu, T. K. F., Xia, Q., Zhou, X., Chai, C. S., & Cheng, M. (2023). Systematic literature review on opportunities, challenges, and future research recommendations of artificial intelligence in education. Computers and Education: Artificial Intelligence, 4, 100118. https://doi.org/10.1016/j.caeai.2022.100118
Zawacki-Richter, O., Marín, V. I., Bond, M., & Gouverneur, F. (2019). Systematic review of research on artificial intelligence applications in higher education – where are the educators? International Journal of Educational Technology in Higher Education, 16(1), Article 39. https://doi.org/10.1186/s41239-019-0171-0
These additions ensure our manuscript is grounded in a more diverse and international body of literature, moving beyond a narrow geographic focus. The cited works strengthen the theoretical foundation by incorporating Western, comparative, and critical perspectives on AI in education, thereby enhancing the global relevance and scholarly rigor of our study.
Comment 15: Revise reporting: Bring full information into tables for statistical statistics (df, F, and p, η2) and consistency in the allocation of abbreviations.
Response 15
We thank the reviewer for the suggestion to improve statistical reporting. We have comprehensively revised all tables to include complete degrees of freedom (e.g., 3, 1098), standardized *p*-value reporting (e.g., p < .001), and added descriptive titles and notes. Please see the updated Tables 2, and 3 in the manuscript (page 8).
For highlighting the need for greater consistency in abbreviations, in response, we have conducted a thorough check and standardization of all abbreviations throughout the manuscript. Specifically, the abbreviation for Willingness to Communicate with AI has been consistently formatted as “AI-WTC” in all relevant sections, particularly in the Results and Tables, replacing any inconsistent usage. Additionally, in the Appendix, we have corrected the conflation of ‘Perceived Usefulness’ with the ‘PEU’ acronym, ensuring it now correctly and consistently represents ‘Perceived Ease of Use’. These revisions ensure terminological precision and uniformity across the entire document.
Round 2
Reviewer 2 Report
Comments and Suggestions for Authors
Thank you revisiting the manuscript in light of my comments. The updated work is more coherent and valid in its procedures and findings. I appreciate the much-improved nature of this work.
Author Response
Comment: Thank you revisiting the manuscript in light of my comments. The updated work is more coherent and valid in its procedures and findings. I appreciate the much-improved nature of this work.
Response: Thank you for your encouraging feedback. We appreciate your valuable comments that helped improve the manuscript.
Reviewer 4 Report
Comments and Suggestions for Authors
Thank you for the privilege of reviewing your thoughtful and comprehensive. I thought the subject matter was very timely as it relates to what is currently happening in AI-based foreign language learning. Your intertwining of the TAM, WTC and sociocultural theory does present a novel perspective that adds value. The manuscript is of high methodological quality and conceptual clarity, and will make an important contribution to Behavioral Sciences.
Having said that, several aspects can be further refined for theoretical depth, readability and practical significance. Please see attached report.
Dear Authors,
Thank you for the privilege of reviewing your thoughtful and comprehensive. I thought the subject matter was very timely as it relates to what is currently happening in AI-based foreign language learning. Your intertwining of the TAM, WTC and sociocultural theory does present a novel perspective that adds value. The manuscript is of high methodological quality and conceptual clarity, and will make an important contribution to Behavioral Sciences.
Having said that, several aspects can be further refined for theoretical depth, readability and practical significance.
Theoretical integration
The sociocultural context is hopeful yet remains superificial. I don't have a suggestion for how to frame it, but how does sociocultural theory (esp mediation and scasflolding & ZPD) influence the connection between scenario context, technology acceptance, and communicative willingness/helpers? This can further support the claim in your argument that AI operates as a “more capable peer.”
Link between TAM and WTC
The two approaches are developed almost independently. For example, clarify if Perceived Ease of Use (PEU) or Perceived Usefulness (PU) mediates AI-WTC or can serve as a predictor of from constructs such as anxiety reduction and social presence. A concept diagram would make this integration clearer.
Discussion and interpretation
The discussion section summarizes the findings, but it could delve deeper to explain why some contexts (especially Factual Information Retrieval) are superior to others. You might frame this in terms of task complexity,communicative risk, or cognitive load, particularly connected with the Chinese educational environment.
Cultural contextualization
Your interpretation connecting low AI-WTC in advisory context with hierarchical teacher–student relationships is thought-provoking but indeed conjectural at this stage. Reinforce this argument with literature on Confucian heritage education, power distance or student silence in East Asian EFL contexts.
Conciseness and readability
The manuscript is thorough, but in some parts also quite long (for example literature review and methods). Try to minimize repeat cites (e.g., multiple references of Fathi et al., Zhang et al) and relegate some of the psychometric information (AVE, it en CR) in an appendix or supplement.
Visualization of results
It would have been useful for readers to view graphical representations (e.g., line or bar graphs) of the mean difference in PU, PEU, and AI-WTC for all four scenarios. This would lend an easier interpretation of the scenario effects.
Future directions
You might also want to propose more elaborate follow-up work, for example involving behavioral data (e.g., chat logs or AI transcripts), cross-cultural studies, or longitudinal designs to investigate change over time in learners’ endorsement and willingness.
Abstract
Simplify and minimize use of acronyms; highlight the key contribution and central findings.
Language
Minor language editing would be helpful to achieve smoother text (e.g., divide long sentences, avoid repetitions of acronyms such as AIS, LSSS etc.).
References
Please ensure that unpublished and in press references (e.g., Peng & Liang, 2025; Rahimi, 2025) have been updated prior to submission.
Formatting
Follow the recommendations of Behavioral Sciences for table placement numbering and alignment.
Ethics and transparency
Great introduction of IRB and AI-use stmt -- keep it for the final version(Objectivity)礼Need to revise.
Overall evaluation
This paper makes an important new contribution to the space by presenting a scenario-sensitive framework for examining AI-mediated communication. It is empirically sound, theoretically ambitious and fits well within the scope of the journal. With moderate changes to streamline the writing and theoretical engagement, it would be ready for publication.
Recommendation
Minor to Moderate Revision
Comments on the Quality of English Language
The writing can be improved. Thank you.
Author Response
Reviewer’s comments and my responses
We would like to sincerely thank the editor and reviewers for their careful reading and constructive feedback on our manuscript. We have carefully considered all comments and revised the paper accordingly to improve its clarity, conciseness, and overall contribution. The following section presents our detailed, point-by-point responses to all the comments.
Comment 1 Theoretical integration
The sociocultural context is hopeful yet remains superificial. I don’t have a suggestion for how to frame it, but how does sociocultural theory (esp mediation and scasflolding & ZPD) influence the connection between scenario context, technology acceptance, and communicative willingness/helpers? This can further support the claim in your argument that AI operates as a “more capable peer.”
Response
We sincerely thank the reviewer for this insightful comment regarding the theoretical integration of sociocultural perspectives. We agree that a deeper engagement with the concepts of scaffolding and ZPD is essential. In response, we have substantially revised the Discussion section (see page 9) to ground our interpretation in the sociocultural concept of AI as a “more capable peer.” We now analyze how the task complexity, communicative risk, and cognitive load inherent in each scenario determine whether the AI’s scaffolding can effectively operate within the learner’s ZPD, thereby influencing technology acceptance and willingness to communicate. For instance, in low-complexity scenarios like Factual Information Retrieval, the AI provides precise, readily integrated scaffolding that minimizes cognitive load and communicative risk, leading to high PU, PEU, and AI-WTC. Conversely, in high-complexity scenarios like Advisory Interaction, the required scaffolding becomes ambiguous and demanding, placing it outside the learner’s ZPD. This mismatch, particularly salient in the context of structured Chinese educational practices, results in high cognitive load, low PEU, and ultimately suppressed AI-WTC, despite potentially high PU. This refined interpretation strengthens our claim by showing that the AI’s efficacy as a peer is not inherent but is co-constructed through the dynamic interplay between the scenario’s design and the learner’s socioculturally shaped perceptions.
Comment 2: Link between TAM and WTC
The two approaches are developed almost independently. For example, clarify if Perceived Ease of Use (PEU) or Perceived Usefulness (PU) mediates AI-WTC or can serve as a predictor of from constructs such as anxiety reduction and social presence. A concept diagram would make this integration clearer.
Response
We appreciate the reviewer’s insightful comment regarding the theoretical integration between Technology Acceptance and Willingness to Communicate. Indeed, prior research has modeled TA as a predictor or mediator of WTC, examining pathways such as perceived usefulness or ease of use leading to reduced anxiety or enhanced social presence, which in turn increase learners’ communicative willingness. While these studies have provided valuable insights, the present research adopts a different yet complementary perspective. In our conceptualization, TA (operationalized as PU and PEU) represents the cognitive-evaluative layer of learner engagement with AI, whereas AI-WTC captures the behavioral-intentional layer of actual communicative readiness. Rather than testing a mediation or prediction model, this study focuses on how distinct interaction scenarios shape these two layers. Specifically, we investigate (1) whether scenario types influence learners’ technology acceptance, (2) whether they also influence their willingness to communicate, and (3) whether the patterns of variation between TA and AI-WTC across scenarios reveal meaningful interconnections. This approach allows us to examine detailed scenario-specific convergence or divergence between cognition (acceptance) and behavior (communication), thereby providing a grounded understanding of when and why cognitive evaluations of AI may, or may not translate into communicative engagement. A conceptual diagram was added in the revised version (Figure 1) to clarify this theoretical structure (see page 3).
Comment 3: Discussion and interpretation
The discussion section summarizes the findings, but it could delve deeper to explain why some contexts (especially Factual Information Retrieval) are superior to others. You might frame this in terms of task complexity, communicative risk, or cognitive load, particularly connected with the Chinese educational environment.
Response
We sincerely thank the reviewer for this constructive suggestion. We have thoroughly revised the Discussion section (see page 9), particularly the third paragraph, to provide a deeper explanation of why certain scenarios, such as Factual Information Retrieval, were perceived more positively than others. Specifically, we have framed our interpretation around the concepts of task complexity, communicative risk, and cognitive load, and connected them explicitly to the Chinese educational environment. We argue that the Factual Information Retrieval scenario’s superiority stems from its low complexity and minimal communicative risk, which align with the structured, accuracy-oriented tasks prevalent in Chinese foreign language classrooms. This alignment reduces cognitive load and allows AI to function effectively as a “more capable peer” within the learner’s ZPD. In contrast, complex scenarios like Advisory Interaction introduce high cognitive load and communicative uncertainty, which conflict with familiar learning schemas and thus suppress willingness to communicate.
We believe these revisions have significantly strengthened the theoretical and contextual grounding of our discussion.
Comment 4: Cultural contextualization
Your interpretation connecting low AI-WTC in advisory context with hierarchical teacher–student relationships is thought-provoking but indeed conjectural at this stage. Reinforce this argument with literature on Confucian heritage education, power distance or student silence in East Asian EFL contexts.
Response
We thank the reviewer for the insightful critique regarding the interpretation of low AI-WTC in advisory contexts. We agree that the initial connection to hierarchical teacher-student relationships, while thought-provoking, was indeed somewhat conjectural without stronger direct evidence. In direct response to this comment, we have removed that specific conjectural interpretation from the discussion section (see page 9). Instead, we have substantially strengthened our argument by reframing the analysis around the more directly observable and theoretically grounded concepts of task complexity, communicative risk, and cognitive load, as also suggested by the reviewer. We have explicitly connected these factors to the nature of learning tasks in the Chinese educational environment, explaining how the high demands and ambiguity of advisory interactions create barriers to communication thereby providing a more robust and less speculative explanation.
Comment 5: Conciseness and readability
The manuscript is thorough, but in some parts also quite long (for example literature review and methods). Try to minimize repeat cites (e.g., multiple references of Fathi et al., Zhang et al) and relegate some of the psychometric information (AVE, it en CR) in an appendix or supplement.
Response
We sincerely appreciate the reviewer’s insightful comment regarding conciseness and readability. In response, we conducted a thorough language and structure review throughout the manuscript to enhance clarity and reduce redundancy. Specifically:
Introduction part: We have shortened the first two paragraphs by removing overlapping explanations about GenAI’s role in L2 learning and the sociocultural perspective. Repetitive phrases and extended theoretical elaborations were condensed to ensure a more focused presentation of the study’s rationale and research gaps (see pages 1-3).
Methods: Several descriptive sentences were streamlined to improve readability. Repeated information about the procedure were deleted or rephrased. We also divided overly long sentences into shorter, more digestible structures to improve flow (see pages 4-7).
Citations: Repeated references (e.g., two mentions of Fathi et al. and Zhang et al. within the same paragraph) were reduced to single citations where appropriate, without losing critical support for key arguments (see page 2).
Appendix: Following the reviewer’s suggestion, we added the detailed psychometric indices (AVE, item loadings, and CR) to Appendix B, allowing the main body to remain concise while keeping the information accessible for reference (see page 15-16).
These revisions collectively improve the manuscript’s readability, coherence, and professional tone while ensuring that all essential methodological transparency is maintained.
Comment 6: Visualization of results
It would have been useful for readers to view graphical representations (e.g., line or bar graphs) of the mean difference in PU, PEU, and AI-WTC for all four scenarios. This would lend an easier interpretation of the scenario effects.
Response
We sincerely thank the reviewer for the valuable suggestion to enhance the presentation of our results. In direct response, we have consolidated the post-hoc comparisons into a streamlined summary (see page 9). This revised format provides a clearer overview of the scenario hierarchies while meticulously preserving statistical accuracy, as it distinctly captures nuanced relationships such as the non-significant difference between LSSS and AIS. We believe that this approach improves readability and conciseness, effectively addressing the reviewer’s objective. The revised comparison results are as follows:
FIRS > AKIS > AIS > LSSS; FIRS > AKIS > LSSS > AIS; FIRS > AKIS > LSSS ≈ AIS
Comment 7: Future directions
You might also want to propose more elaborate follow-up work, for example involving behavioral data (e.g., chat logs or AI transcripts), cross-cultural studies, or longitudinal designs to investigate change over time in learners’ endorsement and willingness.
Response
We sincerely thank the reviewer for this constructive suggestion to elaborate on potential follow-up work. We agree that outlining specific future research directions significantly strengthens the contribution of our study. In direct response, we have revised the conclusion section(see page 11) in the manuscript’s conclusion to propose a more concrete and elaborate agenda. The revised text now explicitly outlines three key avenues: The analysis of behavioral data (e.g., chat logs) to triangulate self-reported perceptions with actual interaction patterns. The implementation of cross-cultural comparative studies to test the generalizability of the scenario effects. The adoption of longitudinal designs to investigate the dynamic evolution of technology acceptance and AI-WTC over time.
We believe these proposed directions provide a clear and valuable roadmap for advancing this line of inquiry, and we are grateful for the reviewer’s guidance in helping us enhance this part of our manuscript.
Comment 8: Abstract
Simplify and minimize use of acronyms; highlight the key contribution and central findings.
Response
We sincerely thank the reviewer for these constructive and valuable suggestions. We have thoroughly revised the manuscript to address the points raised, the changes are detailed below:
We have removed all non-essential acronyms from the abstract, introduction, and discussion sections. The four interaction scenarios are now primarily referred to by their full names (e.g., “Advisory Interaction Scenario”) in these key sections, reserving the acronyms (AIS, LSSS, AKIS, FIRS) only for tables, figures, and the results section where repeated use is necessary for clarity and conciseness. Additionally, we have revised the Discussion section (see page 10) by condensing the responses to the first three research questions and highlighting the study’s key contributions and central findings more explicitly.
Comment 9: Language
Minor language editing would be helpful to achieve smoother text (e.g., divide long sentences, avoid repetitions of acronyms such as AIS, LSSS etc.).
Response
Thank you for the suggestion. We have refined the language throughout the manuscript to improve fluency by dividing long sentences and removing unnecessary repetitions of acronyms, particularly those referring to the four interaction scenarios.
Comment 10: References
Please ensure that unpublished and in press references (e.g., Peng & Liang, 2025; Rahimi, 2025) have been updated prior to submission.
Response:
We thank the reviewer for this important reminder. We have meticulously checked all references and updated the publication status of the previously “in press” citations. Specifically, the reference for Rahimi (2025) has been updated in the reference list with its final publication details, including the volume, issue number, and DOI, as shown below (see page 17):
Rahimi, M. (2025). Synergising dialogic teaching with competencies-trained GenAI dialoguing for critical thinking and communication competencies. Journal of University Teaching and Learning Practice, 22(2). https://doi.org/10.53761/vaa8h214
All other references have been verified to ensure they are the final, published versions.
Comment 11: Formatting
Follow the recommendations of Behavioral Sciences for table placement numbering and alignment.
Response
We appreciate the reviewer’s careful attention to formatting details. In response, we have conducted a thorough check and corrected three inconsistencies in table numbering and alignment in the Results section (see pages 6, 7-8). And all tables in the manuscript have now been formatted as three-line tables in accordance with the journal’s guidelines. The manuscript now fully conforms to the table presentation standards of Behavioral Sciences.